# Organizing structural principles of the IL-17 ligand–receptor axis

Steven C. Wilson[1,2,5], Nathanael A. Caveney[1,2,5], Michelle Yen[1,2], Christoph Pollmann[3], Xinyu Xiang[1,2], Kevin M. Jude[1,2,4], Maximillian Hafer[3], Naotaka Tsutsumi[1,2,4], Jacob Piehler[3] & K. Christopher Garcia[1,2,4✉]

The IL-17 family of cytokines and receptors have central roles in host defence against infection and development of inflammatory diseases[1]. The compositions and structures of functional IL-17 family ligand–receptor signalling assemblies remain unclear. IL-17E (also known as IL-25) is a key regulator of type 2 immune responses and driver of inflammatory diseases, such as allergic asthma, and requires both IL-17 receptor A (IL-17RA) and IL-17RB to elicit functional responses[2]. Here we studied IL-25–IL-17RB binary and IL-25–IL-17RB–IL-17RA ternary complexes using a combination of cryo-electron microscopy, single-molecule imaging and cell-based signalling approaches. The IL-25–IL-17RB–IL-17RA ternary signalling assembly is a C2-symmetric complex in which the IL-25–IL-17RB homodimer is flanked by two 'wing-like' IL-17RA co-receptors through a 'tip-to-tip' geometry that is the key receptor–receptor interaction required for initiation of signal transduction. IL-25 interacts solely with IL-17RB to allosterically promote the formation of the IL-17RB–IL-17RA tip-to-tip interface. The resulting large separation between the receptors at the membrane-proximal level may reflect proximity constraints imposed by the intracellular domains for signalling. Cryo-electron microscopy structures of IL-17A–IL-17RA and IL-17A–IL-17RA–IL-17RC complexes reveal that this tip-to-tip architecture is a key organizing principle of the IL-17 receptor family. Furthermore, these studies reveal dual actions for IL-17RA sharing among IL-17 cytokine complexes, by either directly engaging IL-17 cytokines or alternatively functioning as a co-receptor.

Cytokines of the IL-17 family exert numerous regulatory functions in the immune system and are highly potent pro-inflammatory mediators. IL-17 cytokines are, on the one hand, essential for barrier tissue homeostasis, but on the other hand act as drivers of various autoimmune disorders[1]. Activation of IL-17-induced signalling cascades leads to induction of chemokines at sites of inflammation. The IL-17 family of cystine-knot cytokines consists of six members: IL-17A–F, which exhibit differential binding specificities to five receptors, IL-17RA–E. The most well-studied IL-17 cytokines, IL-17A and IL-17F, are produced by T helper 17 ($T_H17$) cells and are elevated in the context of inflammation[3–7]. As such, monoclonal antibodies that inhibit the activity of IL-17A and IL-17RA, the receptor for IL-17A and IL-17F, are US FDA-approved drugs for plaque psoriasis and other autoimmune disorders[8]. Another family member, IL-25, is a key regulator of type 2 immunity[9–13] and is an important therapeutic target for $T_H2$-driven disorders including allergic asthma and atopic dermatitis[14].

Unambiguous ligand–receptor matching and the compositions of active ligand–receptor signalling assemblies in this family have been difficult to discern due to varied use of both shared (IL-17RA) and ligand-specific receptor subunits. Both IL-17A and IL-17F require IL-17RA and IL-17RC[15–17] for signalling, whereas IL-25 binds only to IL-17RB[18] yet requires both IL-17RA and IL-17RB (Fig. 1a) for signalling in vivo[2]. IL-17A, IL-17F and IL-25 signal via the adaptor protein and E3 ubiquitin ligase ACT1 (refs. [19–21]), which interacts with IL-17R via a homotypic interaction with the SEF/IL-17R (SEFIR) domain of IL-17R[22]. IL-17RA is proposed to act as a common IL-17R in heterodimeric complexes with most other IL-17R[1,23]. Structures have been reported for incomplete 1:1 complexes of IL-17RA with IL-17F[24], the IL-17A–IL-17F heterodimer[25] and IL-17A[26], respectively, where the cytokines leave an 'open face' unoccupied for a presumed second receptor to bind to form a 1:2 ligand:receptor signalling complex. There is also a structure of IL-17F bound to two IL-17RC in a homodimeric complex[23] whose signalling competence remains unclear. On the one hand, cellular responses to IL-17A and IL-17F signalling have been shown to require IL-17RA in mice[16] and humans[27]; on the other hand, IL-17F has been shown to upregulate IL-33 in mouse cells lacking IL-17RA[28]. Thus, the mechanistic basis for how extracellular engagement of the IL-17 family of cytokines with their receptors leads to signalling remains to be structurally defined.

We elucidated the composition and structure of complete heterodimeric IL-17–IL-17R signalling complexes using the complementary approaches of cryo-electron microscopy (cryo-EM), single-molecule

[1]Department of Molecular and Cellular Physiology, Stanford University School of Medicine, Stanford, CA, USA. [2]Department of Structural Biology, Stanford University School of Medicine, Stanford, CA, USA. [3]Divison of Biophysics, Department of Biology, University of Osnabrück, Osnabrück, Germany. [4]Howard Hughes Medical Institute, Stanford University School of Medicine, Stanford, CA, USA. [5]These authors contributed equally: Steven C. Wilson, Nathanael A. Caveney. ✉e-mail: kcgarcia@stanford.edu

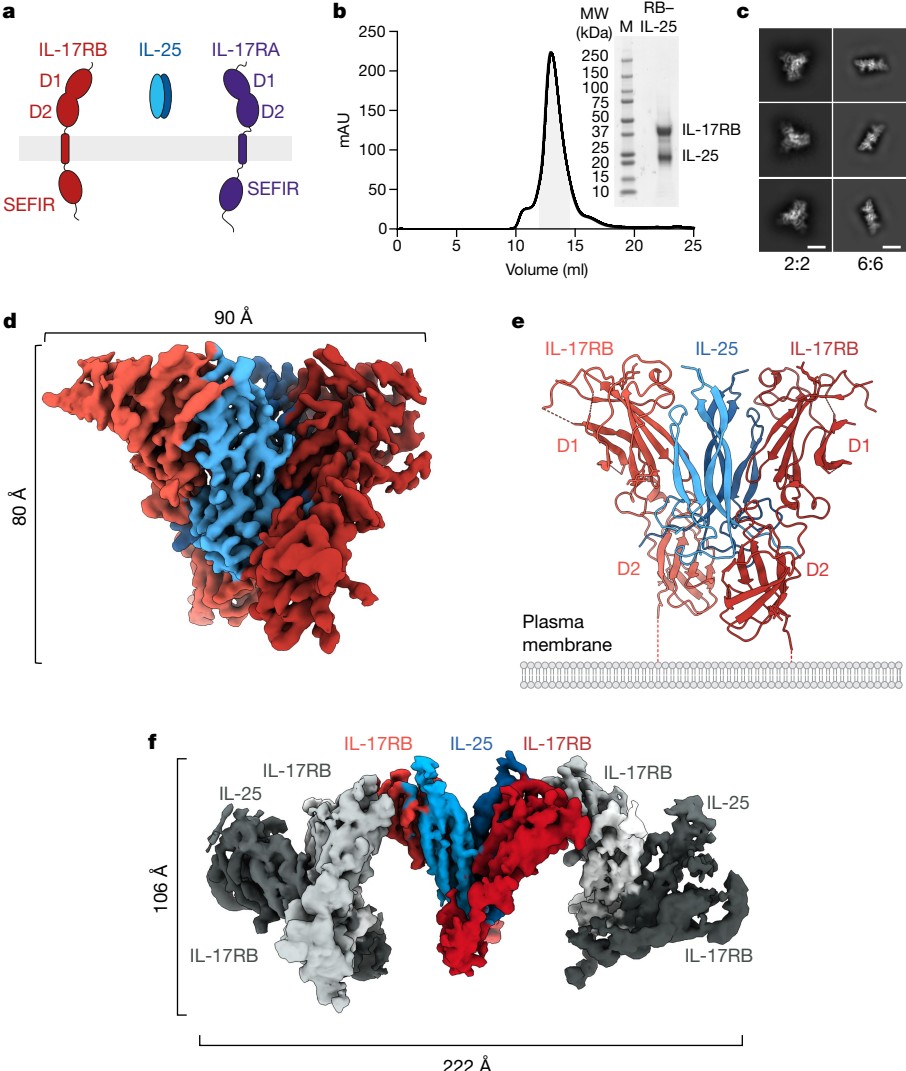

**Fig. 1 | Cryo-EM structures of binary and higher-order IL-25–IL-17RB complexes. a**, Schematic showing the components required for IL-25 signalling: IL-17RB, IL-25 and IL-17RA. **b**, Gel filtration chromatogram and SDS–PAGE gel image for the IL-25–IL-17RB complex (RB–IL-25). The highlighted area under the size-exclusion chromatography curve represents the pooled fractions that were concentrated and vitrified for cryo-EM. MW, molecular weight. **c**, Two-dimensional classifications of binary (2:2) IL-25–IL-17RB

homodimeric complexes and higher-order IL-25–IL-17RB (6:6) complexes. Scale bars, 55 Å (2:2 2D class averages) and 125 Å (6:6 2D class averages). **d**, Cryo-EM map of the 2:2 IL-25–IL-17RB complex. **e**, Model of the 2:2 IL-25–IL-17RB complex. Dashed lines indicate unmodelled linkers to the transmembrane regions of the receptors. **f**, Cryo-EM map of the higher-order IL-25–IL-17RB (6:6) complex, comprising a trimer of IL-25–IL-17RB homodimers.

imaging and cell-based structure–function signalling experiments. In the cryo-EM structures of IL-25–IL-17RB non-signalling and IL-25–IL-17RB–IL-17RA signalling complexes, IL-25 does not form the expected heterodimeric complex with IL-17RB and IL-17RA. Instead, IL-25 forms a homodimeric complex with IL-17RB, which recruits the binding of IL-17RB to IL-17RA in a heteromeric 2:2:2 ternary complex in which the receptors interact via their membrane-distal tips. Using these structures, we performed structure-guided functional studies to demonstrate that IL-25 allosterically induces the formation of this IL-25–IL-17RB–IL-17RA ternary complex required for IL-25 signalling. We further conducted cryo-EM studies of additional IL-17–IL-17R complexes to determine the generality of this unusual architecture and found that the tip-to-tip receptor–receptor mode of assembly extends to other IL-17–IL-17R. Thus, IL-17RA serves as an organizing hub mediating IL-17 family signalling at the membrane by acting either as a direct receptor for IL-17 ligands or as a co-receptor.

## Structures of IL-25–IL-17RB complexes

We co-expressed IL-25 and IL-17RB in Expi293F GnTI⁻ cells, and they eluted as a stable complex over gel filtration (Fig. 1b). Two-dimensional classification of the particles in this sample revealed symmetric bi-lobed structures (Fig. 1c), consistent with a 2:2 homodimeric receptor complex. Using a tilted collection strategy, we determined the structure of the IL-25–IL-17RB complex using single-particle cryo-EM to a global resolution of 3.18 Å (Fig. 1d, Extended Data Table 1 and Extended Data Figs. 1 and 2a,b). In the structure, IL-25 forms a homodimeric 2:2 complex with IL-17RB, in which the centrally located IL-25 engages two copies of IL-17RB at its two receptor-binding faces (Fig. 1d,e).

During 2D classification of the IL-25–IL-17RB dataset, we observed classes of higher-order IL-25–IL-17RB structures (Fig. 1c). Further classification and refinement revealed arrayed trimers of 2:2 IL-25–IL-17RB modules. We refined these particles to produce a map with a global resolution of 4.39 Å (Fig. 1f, Extended Data Table 1 and Extended Data Fig. 3). The higher-order IL-25–IL-17RB complex consists of a core IL-25–IL-17RB

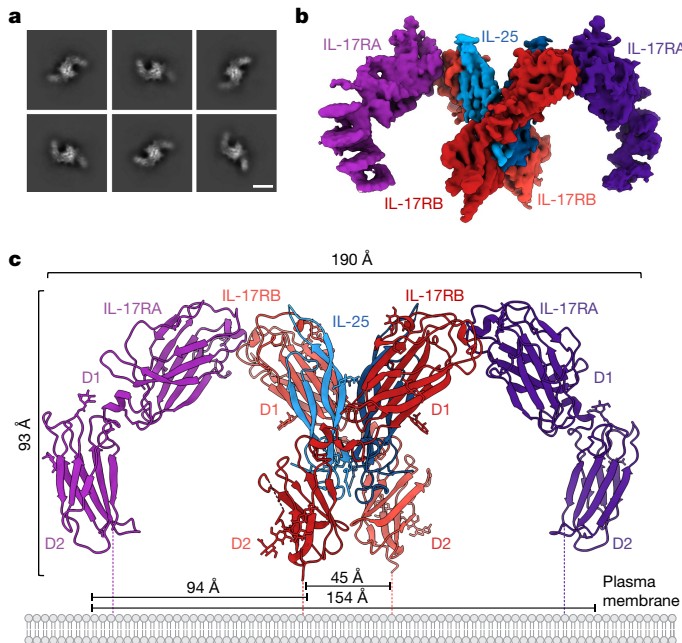

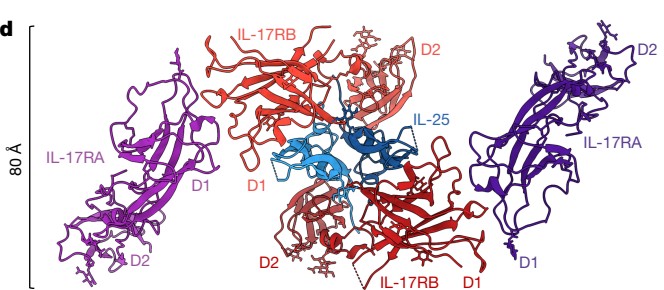

**Fig. 2 | Cryo-EM structure and architecture of the IL-25–IL-17RB–IL-17RA ternary complex. a**, 2D classifications of ternary IL-25–IL-17RB–IL-17RA complexes. Scale bar, 82 Å (2D class averages). **b**, Side view of the C2-symmetric cryo-EM map of the IL-25–IL-17RB–IL-17RA ternary complex reveals IL-17RA wings emanating from a core IL-25–IL-17RB complex. **c**, Side view of the ternary IL-25–IL-17RB–IL-17RA complex model shows that the D1 domain of IL-17RA interacts exclusively with the D1 domain of IL-17RB. Dashed lines indicate unmodelled linkers to the transmembrane regions of the receptors. Distances between the centres of masses of the D2 domains are shown. **d**, Top view of the IL-25–IL-17RB–IL-17RA complex model.

homodimer flanked by a homodimer on each side. Although there is, in principle, the potential for continued polymerization beyond this arrayed formation, we did not see classes beyond three homodimers. These IL-25–IL-17RB homodimers are connected by interactions between the membrane-distal tips of the IL-17RB type III fibronectin D1 domains (Fig. 1f). The physiological relevance of this higher-order trimer of homodimers is not clear, but given that IL-17RB alone is not able to support signalling, we speculate that this is not a signalling active assembly.

## Structure of the IL-25 ternary complex

With the goal of obtaining the IL-25–IL-17RB–IL-17RA ternary complex structure, we mixed excess purified IL-17RA with purified IL-25–IL-17RB before vitrification for cryo-EM (Extended Data Fig. 4e). During initial 2D classification, we observed classes of particles that appeared to be the IL-25–IL-17RB homodimer with IL-17RA 'wings' (Fig. 2a). Data collection and refinement resulted in a C2-symmetric map with a global resolution of 3.66 Å (Fig. 2b, Extended Data Table 1 and Extended

Data Figs. 2c,d and 4) and confirmed that these particles were indeed IL-25–IL-17RB–IL-17RA ternary complexes (Fig. 2c,d and Extended Data Fig. 2e).

The ternary complex consists of a 2:2 IL-25–IL-17RB homodimeric core flanked by two copies of IL-17RA (Fig. 2c,d). Notably, the IL-17RA molecules do not directly contact IL-25, interacting solely with IL-17RB (Figs. 2c,d and 3a,f). The D1 domains of IL-17RA and IL-17RB interact at their membrane-distal tips through highly complementary surfaces (Figs. 2c,d and 3a,f). In this geometry, the membrane-proximal regions of IL-17RA and IL-17RB are separated by approximately 94 Å, rather than being in close proximity as seen in other classes of dimeric signalling receptor such as cytokine receptors and RTKs.

## IL-25 signals via IL-17RB binding

There are two disparate ligand–receptor and receptor–receptor interfaces mediating the complex assembly: IL-25–IL-17RB and IL-17RB–IL-17RA. We interrogated the importance of each interface for IL-25-induced signalling. The 2:2 IL-25–IL-17RB complex contains three distinct IL-25–IL-17RB interaction sites per IL-25–IL-17RB dimer for a total of six sites (Fig. 3a–e). Site 1 (buried surface area (BSA) = 499 Å$^2$) is near the membrane-distal tip of the complex and contains IL-17RB D1 domain α1, β7 and β8, and loops 1 and 9, and IL-25 β1 and β2, and loops 2 and 4 (Fig. 3a). Within site 1, IL-17RB W34 of α1 forms hydrophobic pockets with loop 1 and loop 9 on its sides, into which L98 and L101 of IL-25 loop 2 insert (Fig. 3b). Site 2 (BSA = 636 Å$^2$), near the bottom of the IL-17RB D1 domain, is composed of IL-17RB D1 β4, β5 and β7, and loops 6 and 8 and a composite interface made of the N terminus, $3_{10}$ helix, β1, β2, β4 of one IL-25 chain and β3 and β4 of the other IL-25 chain (Fig. 3a). Key site 2 residue–residue interactions include F135 of IL-17RB β7 forming a pi-stacking interaction with Y106 of IL-25 β2 in the first chain (Fig. 3c), and Y94 at the tip of IL-17RB loop 6 burying into a groove formed by Y134 and N136-NAG on β3 of IL-25 in the second chain (Fig. 3d). Site 3 (BSA = 760 Å$^2$) is formed by the helical linker and IL-17RB D2 domain loops 10, 11, 12, 15 and 18, and IL-25 loop 3 and the C terminus of the first chain, and the N terminus, $3_{10}$ helix 2 and loop 3 of the second chain (Fig. 3a). Within the interface, the N terminus of one IL-25 chain makes contacts with the helical linker of IL-17RB (Fig. 3e), whereas the C terminus of the other IL-25 chain threads through a crevice between the helical linker and the D2 domain of IL-17RB (Fig. 3e).

Visual inspection and PISA analysis[29] were used to select several key IL-25 residues (Y92, L98, L101, Y106, Y134 and M176) for alanine substitution to validate the contact interfaces between IL-25 and IL-17RB on cells expressing IL-17RA and IL-17RB (Fig. 3b–e). The IL-25 mutants had weakened ability to drive gene expression in an ACT1-expressing NF-κB reporter cell line, as reflected by increased half-maximal effective concentration (EC$_{50}$) values in comparison to wild-type (WT) IL-25 (Fig. 3g). One mutant, the site 2 mutant Y92A, had substantially reduced efficacy with an EC$_{50}$ shift of more than 3 log-fold relative to WT (Fig. 3g). The site 2 and site 3 mutants also had impaired ability to stimulate type 2 cytokine secretion on human peripheral blood mononuclear cells (Fig. 3h).

## IL-17RB–IL-17RA mediates signalling

To interrogate the role of the IL-17RA–IL-17RB interaction in IL-25 signalling, we generated *IL17RA* CRISPR knockout cell lines carrying an NF-κB-inducible reporter and asked whether the *IL17RA* knockouts (which retain endogenous IL-17RB expression) could respond to IL-25. The *IL17RA* knockouts had a complete loss of signalling when stimulated with IL-25 (Fig. 3i). We also asked whether IL-17RA was able to activate signalling in the absence of IL-17RB by generating *IL17RB*-knockout reporter cell lines and observed that they too lost the ability to respond to IL-25 (Fig. 3i). These results provide strong genetic evidence that both IL-17RB and IL-17RA are essential for IL-25 signal transduction, independently confirming earlier findings[2].

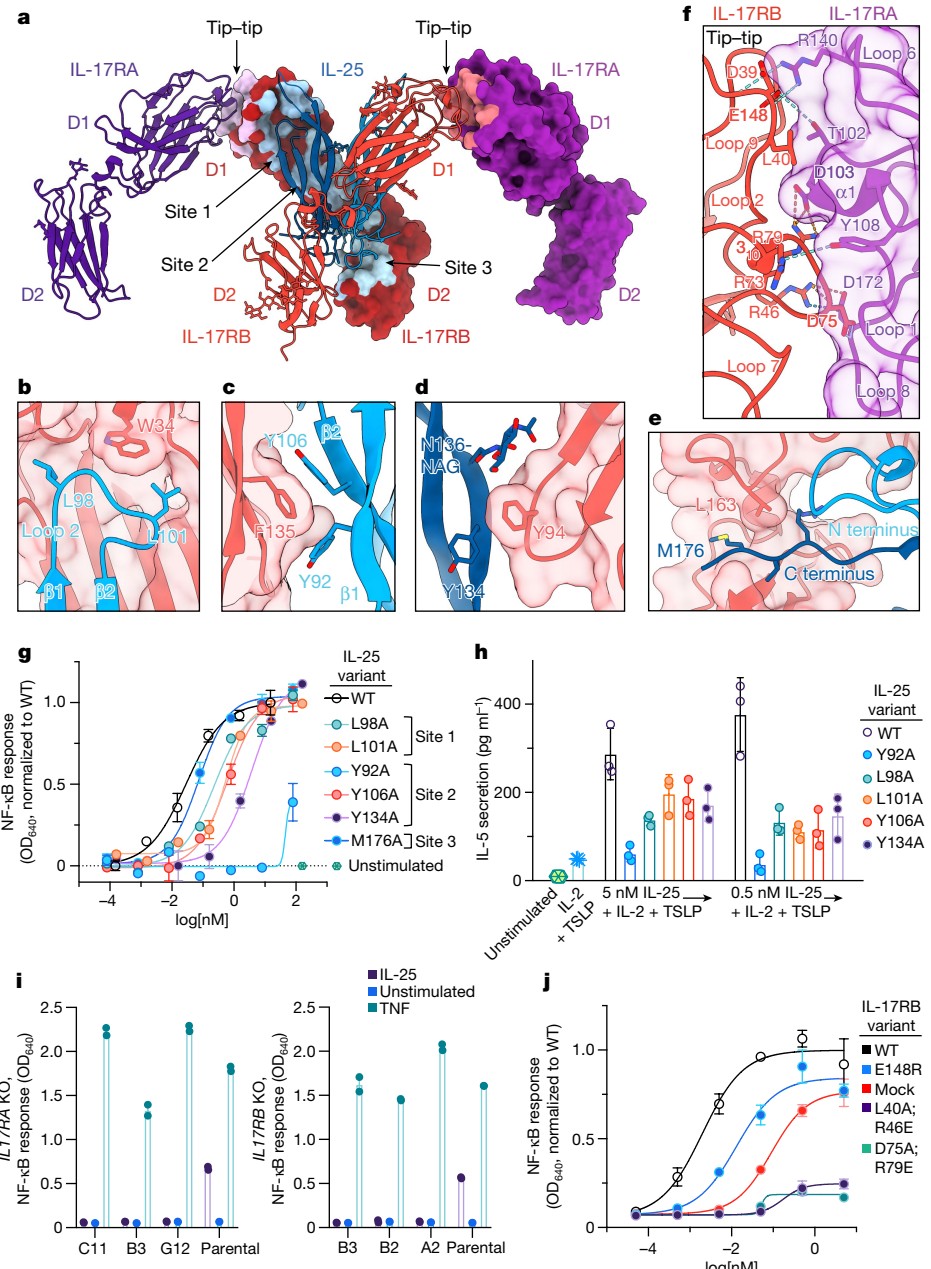

**Fig. 3 | Structure-guided functional studies of the IL-25–IL-17RB–IL-17RA ternary complex. a**, Key interaction sites in the ternary complex are highlighted on the surfaces of IL-17RB and IL-17RA. **b–e**, Key elements involved in IL-25–IL-17RB site 1 (**b**), site 2a (**c**), site 2b (**d**) and site 3 (**e**) are shown. The IL-17RB surface is depicted in light red. NAG, *N*-acetylglucosamine. **f**, Detailed view of an IL-17RA–IL-17RB tip-to-tip interface shows key secondary structures, residues and bonds. The IL-17RA surface is depicted in light purple. **g**, IL-25 mutants have reduced signalling efficacy. Residues were selected for mutagenesis based on their key roles in IL-25–IL-17RB interactions, shown in **b–e**. NF-κB reporter HEK293 cells were stimulated with varying concentrations of WT IL-25 or mutant, then assayed for transcriptional activity. OD, optical density. Dashed grey line depicts normalized baseline. **h**, IL-25 point mutants from **g** have impaired ability to induce IL-5 secretion from peripheral blood mononuclear cells. IL-2, interleukin-2; TSLP, thymic stromal lymphopoietin. **i**, Genetic deletion of the gene encoding IL-17RA (left) or IL-17RB (right) eliminates IL-25-induced NF-κB transcriptional activity. Knockouts (KO) were generated in the background of the NF-κB HEK293 transcriptional reporter line used in **g** and **j**. C11, B3, G12, B2 and A2 represent different clones. **j**, NF-κB HEK293 reporter cells (with IL-17RA and endogenous IL-17RB) were transfected with exogenous WT IL-17RB or tip mutants, then assayed for transcriptional activity in response to IL-25 stimulation. The double mutants show greatly reduced signalling efficacy. The tip mutant design was based on analysis of the interface in **f**. Data in **g**, **h** and **j** represent single experiments from three independent biological experiments, whereas data in **i** are from experiments conducted once. Data are mean ± s.d. with *n* = 2 for **g**, **i** and **j**, and *n* = 3 for **h**.

## IL-25 signals via tip-to-tip interface

IL-17RA interacts exclusively with IL-17RB, rather than IL-25, therefore acting more in the manner of a co-receptor than a direct cytokine receptor (Figs. 2c,d and 3a,f). The IL-17RA–IL-17RB interface (BSA = 570 Å²) is mediated by complementary surfaces composed of IL-17RA α1 and loops 1, 3, 6 and 8, and IL-17RB 3₁₀ helix 1 and loops 2, 4, 7 and 9 (Fig. 3a,f). The interface is stabilized by core hydrophobic interactions near the centre, with a primary interaction mediated by IL-17RA α1 burying into a groove formed by 3₁₀ helix 1 and loops 2 and 9 of IL-17RB (Fig. 3a,f). Similarly, the 3₁₀ helix 1–loop 4 of IL-17RB is buried into a complementary groove formed

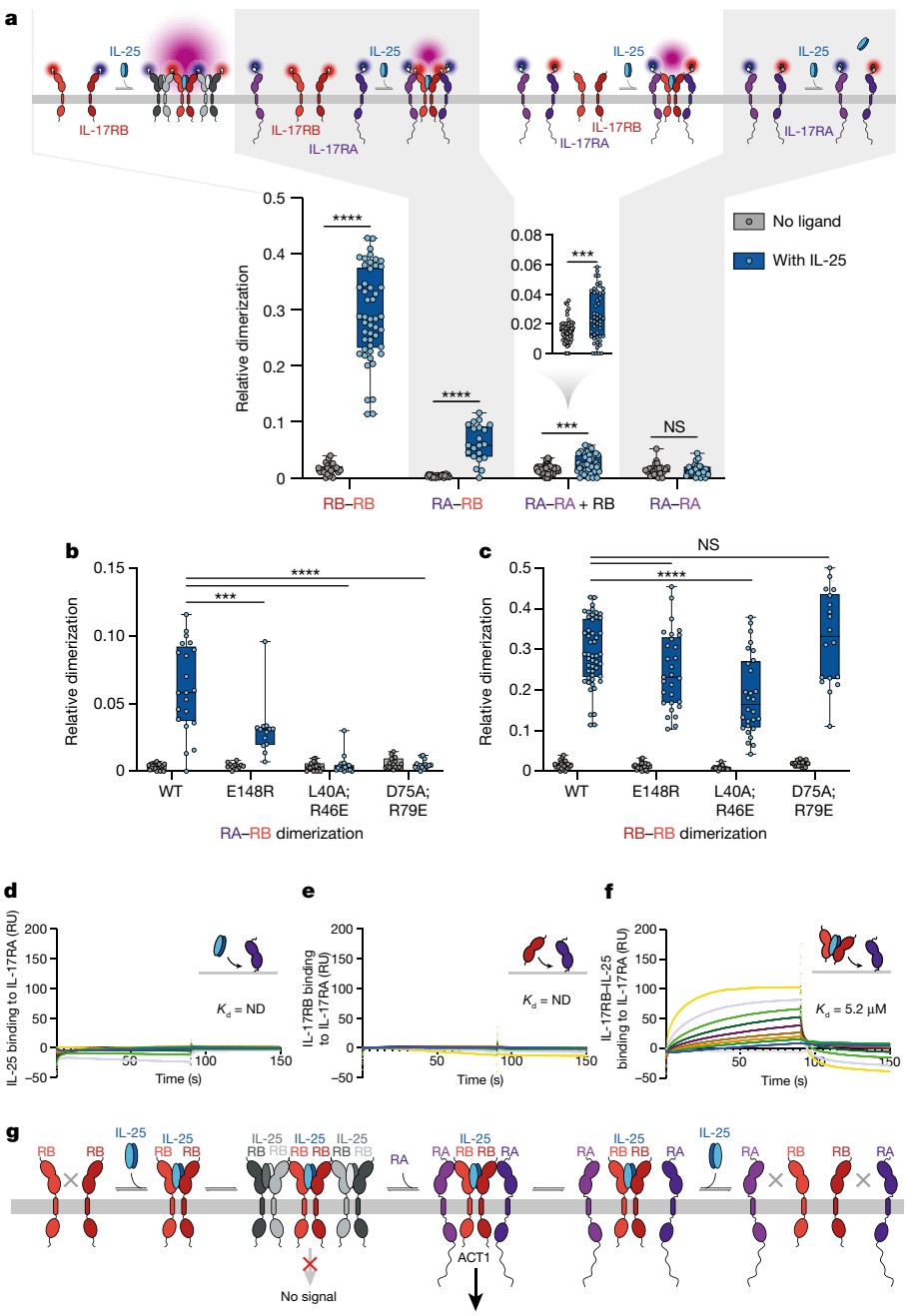

**Fig. 4 | Quantitative interaction analysis in vitro and in live cells reveals IL-25-induced IL-17RB–IL-17RA binding via the tip-to-tip interface. a–c,** IL-17RB–IL-17RA interactions in the plasma membrane quantified by live-cell single-molecule imaging. **a,** Different receptor labelling schemes used to interrogate IL-17RB (RB) and/or IL-17RA (RA) homodimerization and heterodimerization by single-molecule co-tracking (top). Box and whisker plots show relative dimerization levels determined by single-molecule co-tracking for homodimerization and heterodimerization of IL-17RA and IL-17RB in the absence and presence of IL-25 (bottom). Numbers of cells and experiments, respectively, are 25 and 4, 49 and 4, 22 and 2, 21 and 2, 51 and 5, 48 and 5, 23 and 2, and 24 and 2 for each column, going from left to right. **b,c,** Heterodimerization of IL-17RA–IL-17RB (**b**) and homodimerization of IL-17RB (**c**) compared with IL-17RB mutations in the IL-17RA–IL-17RB interface. Numbers of cells and experiments, respectively, are 22 and 2, 21 and 2, 12 and 1,

13 and 1, 19 and 2, 18 and 2, 18 and 1, and 17 and 1 for each column in **b**, and 25 and 4, 49 and 4, 26 and 2, 29 and 2, 17 and 2, 28 and 2, 18 and 1, and 18 and 1 for each column in **c**, going from left to right. Statistics for **a–c** were performed using two-sample Kolmogorov–Smirnov tests (not significant (NS); ***$P$ < 0.001 and ****$P$ < 0.0001). Box and whisker plots in **a–c** show the five number summaries of the data: minimum, first quartile, median, third quartile, and maximum values. **d–f,** SPR sensorgrams reveal that IL-17RB complexes with IL-25 bind to IL-17RA. The analyte concentration range in **d** is a twofold serial dilution ranging from 20 to 0.156 μM. The analyte concentration ranges in **e** and **f** are twofold serial dilutions from 40 to 0.156 μM. Dissociation constants ($K_d$) are indicated on the sensorgrams. SPR sensorgrams represent a single experiment ($n$ = 1) for **d** and one of two independent experiments ($n$ = 2) for **e** and **f**. ND, not determined; RU, response unit. **g,** Model of IL-25–IL-17RB–IL-17RA signalling pathways as inferred from our structural, signalling and biophysical data.

by IL-17RA α1 and loops 1 and 8 (Fig. 3a, f). The IL-17RA–IL-17RB interface is further stabilized by an extensive network of hydrogen bonds and salt bridges (Fig. 3f).

Key interface residues were selected for mutagenesis based on visual inspection and PISA analysis. We generated IL-17RB double mutants L40A;R46E and D75A;R79E because these residues form the core of the

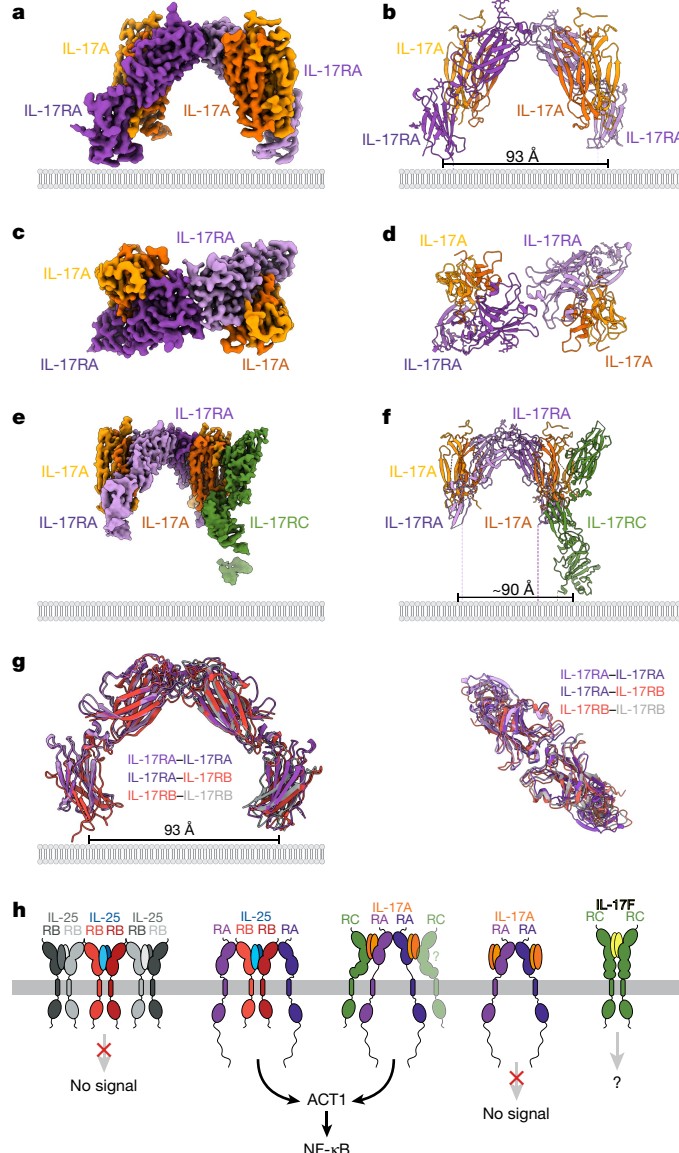

**Fig. 5 | The tip-to-tip architecture is conserved in cryo-EM structures of IL-17A–IL-17RA and IL-17RA–IL-17A–IL-17RC complexes. a,b**, Side views of the map (**a**) and model (**b**) of the IL-17A–IL-17RA tip-to-tip dimer. The structure consists of IL-17–IL-17RA complexes dimerized with each other via an IL-17RA–IL-17RA tip-to-tip interface. **c,d**, Top views of the map (**c**) and model (**d**) of IL-17–IL-17RA in **a** and **b**. **e,f**, Side views of the map (**e**) and model (**f**) of an IL-17A–IL-17RA complex bound to an IL-17RA–IL-17A–IL-17RC heterodimer via an IL-17RA–IL-17RA tip-to-tip interface. **g**, Superposition of the IL-17A–IL-17RA tip-to-tip dimer with the tip-to-tip components of the IL-25–IL-17RB–IL-17RA complex and higher-order IL-17RB–IL-17RB arrays reveals that the tip-to-tip architecture is conserved across IL-17–IL-17R complexes. Dashed lines indicate unmodelled linkers to the transmembrane regions of the receptors. Distances between the centres of masses of the membrane-proximal domains are shown in **b**, **f** and **g**. **h**, Signalling model of a shared IL-17–IL-17R mechanism in which the spacing of IL-17RA is important for transducing IL-17 signals. RC, IL-17RC.

interface and/or the salt bridges on the interface periphery (Fig. 3f), and we also produced the single E148R charge-reversal mutant, which we predicted would disrupt an $R140_{RA}$–$E148_{RB}$ salt bridge at the periphery of the complex (Fig. 3f). In NF-κB transcriptional reporter cells transfected with WT or mutant IL-17RB (Fig. 3j), exogenous WT IL-17RB enhanced maximal effect ($E_{max}$) over that supported by endogenous levels of IL-17RB, whereas the E148R single mutant left $E_{max}$ unchanged,

and the double mutants greatly suppressed it (exogenous WT > E148R ≈ mock transfected > double mutants). Furthermore, the rank order of NF-κB activity $E_{max}$ was supported with a concurrent shift in $EC_{50}$ values, with the rank order of efficacy being: exogenous WT > E148R > mock transfected > double mutants (Fig. 3j).

## Single-molecule imaging of complexes

To understand the relevance of the binary and ternary cryo-EM structures within the cellular environment, we quantified homodimerization and heterodimerization of IL-17RB and IL-17RA on the plasma membrane of live cells. We conducted two-colour single-molecule co-tracking of fluorescently labelled IL-17RA and IL-17RB using total internal reflection fluorescence microscopy (Fig. 4a and Extended Data Fig. 5a–d). These studies revealed that IL-25 induces the formation of IL-17RB homodimers independently of IL-17RA, in line with the stable IL-25–IL-17RB dimers found in vitro. Single-molecule intensity analyses suggested that higher-order clusters are formed as observed in vitro, yet at a rather low level (less than 10% of the co-diffusing fraction; Extended Data Fig. 5e). This low level of higher-order clustering is consistent with the values seen in our cryo-EM samples, in which approximately 18% of particles form higher-order clusters. Likewise, we observed robust IL-25-induced formation of IL-17RA–IL-17RB complexes (Fig. 4a), although at substantially lower levels than IL-17RB homodimerization. This result is in accord with our structural model of the ternary complex and the relatively low affinity of the IL-17RA interaction with the IL-25–IL-17RB complex. In the presence of IL-17RB, IL-25 also weakly stimulated homodimerization of IL-17RA–IL-17RA in agreement with the 2:2:2 complex observed in our structure (Fig. 4a). Concomitantly, no change in IL-25-induced homodimerization of IL-17RB was observed in the presence of IL-17RA (Extended Data Fig. 5f). These results support our model that IL-17RA is recruited into the IL-25–IL-17RB complex, rather than competing for direct IL-25 binding. A minor increase in the immobile fraction in conjunction with graded decrease in the diffusion constants (Extended Data Fig. 5g,h) support the formation of well-defined complexes with stoichiometries as predicted by our structural studies[30].

We also used single-molecule co-tracking to assess the ability of the IL-25 signalling deficient IL-17RB mutants (L40A;R46E, D75A;R79E and E148R) we assayed (Fig. 3j) to associate with each other and with IL-17RA. Consistent with the cell signalling experiments, we observed significantly decreased cytokine-induced heterodimerization of WT IL-17RA with mutant IL-17RB, with the L40A;R46E and D75A;R79E double mutants eliminating IL-25–IL-17RA–IL-17B complex formation, whereas the E148R single mutant retained a low level of IL-17RB–IL-17RA association (Fig. 4b). In contrast to their large effect on IL-17RA–IL-17RB heterodimerization, the E148R and D75A;R79E IL-17RB mutants had minor or undetectable effects on cytokine-induced IL-17RB–IL-17RB homodimerization, whereas the L40A;R46E double mutant had a significant reduction in self-association (Fig. 4c). These results highlight the relevance of the predicted IL-17RA–IL-17RB interface for the recruitment of IL-17RA by IL-25-induced IL-17RB–IL-17RB homodimerization. Reduced IL-17RB dimerization of the L40A;R46E mutant could be explained by these residues contributing to the formation of the higher-order IL-25–IL-17RB arrays.

## IL-25 primes IL-17RB for IL-17RA binding

Together with our cryo-EM structures and mutant signalling data, our single-molecule tracking data support an assembly model in which the binding of IL-25 to IL-17RB induces association of IL-17RB with IL-17RA, as well as the formation of higher-order IL-17RB oligomers (Fig. 4a–c). To test this model directly, we conducted surface plasmon resonance (SPR) binding studies to assess whether IL-25 binding to IL-17RB enhances the affinity of IL-17RB for IL-17RA or IL-17RB. We found that IL-25 enables

IL-17RB binding to immobilized IL-17RA (Fig. 4f), as IL-17RB alone did not bind to immobilized IL-17RA (Fig. 4e). In addition, IL-25 did not interact directly with IL-17RA (Fig. 4d), consistent with the lack of IL-25-induced IL-17RA–IL-17RA association in our single-molecule tracking data (Fig. 4a). We also found that the IL-25-bound IL-17RB complex, but not IL-17RB alone, could bind to immobilized IL-17RB (Extended Data Fig. 6f,g). However, we did observe weak interactions of *apo*-IL-17RA with immobilized IL-17RB (Extended Data Fig. 6j), albeit at a lower affinity (13.8 μM) than for the IL-25–IL-17RB complex binding to immobilized IL-17RA (5.2 μM). Despite the weak *apo*-IL-17RB–*apo*-IL-17RA interaction detected by SPR, we did not detect an IL-25-independent IL-17RB–IL-17RA interaction using single-molecule live-cell imaging (Fig. 4b). We also performed SPR to assess the effects of the IL-17RB L40A;R46E and D75A;R79E mutants on binding of IL-25–IL-17RB complexes to IL-17RA or IL-17RB. We did not detect binding of these mutant complexes to immobilized IL-17RA (Extended Data Fig. 6b,c) or IL-17RB (Extended Data Fig. 6h,i).

Overall, our structural (Figs. 1, 2 and 3a–f), cell signalling (Fig. 3g–j) and biophysical (Fig. 4a–f) data support a model in which the binding of IL-25 to IL-17RB initiates signalling by indirectly, perhaps allosterically, enhancing the formation of IL-25–IL-17RB–IL-17RA ternary complexes via the IL-17RB–IL-17RA tip-to-tip interface. In principle, the path to assembly of the active ternary complex could proceed through two possible pathways: (1) formation of IL-25–IL-17RB homodimeric, or (2) higher-order IL-25–IL-17RB complexes, followed by recruitment of IL-17RA via the tip-to-tip interfaces to generate the signalling complex (Fig. 4g). These data are consistent with the requirement of both IL-17RA and IL-17RB for IL-25 signalling in vivo[2] and provide a mechanistic explanation for IL-25 signalling through the IL-25–IL-17RB–IL-17RA ternary complex.

## Conserved tip-to-tip architecture

We conducted cryo-EM studies of IL-17A–IL-17RA and IL-17A–IL-17RA–IL-17RC complexes. We co-expressed and co-purified complexes containing IL-17A, IL-17RA and IL-17RC. These molecules co-eluted in a monodisperse peak over gel filtration with IL-17A–IL-17RA having an apparent 1:1 stoichiometry, with IL-17RC being sub-stoichiometric on SDS–PAGE (Extended Data Fig. 7a,b). Using single-particle cryo-EM, we determined from a 2.51 Å resolution 3D reconstruction that IL-17A–IL-17RA dimers interacted at the tips of the IL-17RA molecules (Fig. 5a–d, Extended Data Table 1 and Extended Data Fig. 7c–f). We also determined a 3D reconstruction for a minor population of classes containing the IL-17A–IL-17RA dimers with a single IL-17RC bound to IL-17A to a resolution of 3.01 Å (Fig. 5e,f, Extended Data Table 1 and Extended Data Fig. 7c–f); this structure contains a single copy of the complete IL-17RA–IL-17A–IL-17RC heterodimer. The tip-to-tip interface present in the IL-17A–IL-17RA and IL-17A–IL-17RA–IL-17RC structures superimposes with the IL-17RB–IL-17RA tip-to-tip interface in the IL-25–IL-17RB–IL-17RA ternary complex and the IL-17RB–IL-17RB tip-to-tip interface in the higher-order IL-25–IL-17RB complex (Fig. 5g), revealing a large conserved spacing between the IL-17RA–IL-17RA and IL-17RA–IL-17RB membrane-proximal domains, which has likely implications for the assembly of intracellular ACT1 molecules that can support downstream signalling.

## Discussion

This study offers several conclusions that give further insight into the receptor biology of the IL-17 family. First, IL-17RA possesses the very unusual capacity to act as a co-receptor that does not engage cytokine in the IL-25 complex, or in the more traditional role as a directly engaging cytokine receptor, as we have seen in the IL-17A and IL-17F complexes. Second, our results further support the notion that IL-17RA is a central signalling hub of the family and is required for signalling by most known IL-17 family members. The use of shared signalling receptors is also a feature of JAK–STAT cytokines in which, for example, the common gamma chain serves an analogous role as IL-17RA. However, in contrast to the common gamma chain, which is paired as a heterodimer with the ligand-specific receptor subunits, IL-17RA can signal either in the context of direct association with cytokine or act as a co-receptor.

The third finding from our studies is that the tip-to-tip dimer assembly probably represents a structural organizing principle by which IL-17 family members congregate into signalling complexes. The tip-to-tip receptor architecture was present in our cryo-EM structures of an IL-17RA–IL-17RA dimer (Fig. 5a–d) and another IL-17RA–IL-17RA tip-to-tip dimer with IL-17RC bound to one copy of IL-17A (Fig. 5e,f). The tip-to-tip interfaces of these IL-17A-containing structures are superimposable onto our arrayed IL-25–IL-17RB and IL-25–IL-17RB–IL-17RA ternary structures (Fig. 5g). We also surveyed the electron density maps for crystal structures of IL-17RA–IL-17F[24], IL-17RA–IL-17A[26] and IL-17RA–IL-17A/F[25] complexes and found the IL-17RA–IL-17RA tip-to-tip interface in all of the crystals (Extended Data Fig. 8a). On the basis of the ability of IL-25 to enhance IL-17RB binding to IL-17RA, we hypothesize that IL-17A binding stabilizes the membrane-distal tip of IL-17RA and enables IL-17RA–IL-17RA dimerization (Extended Data Fig. 8b). In addition, the tip-to-tip interface is absent from the crystal structure of the homodimeric IL-17F–IL-17RC complex[23], potentially due to the insertion of a negatively charged nine amino acid loop in the membrane-distal tip of IL-17RC (Extended Data Fig. 9).

Binding of IL-25 to IL-17RB and IL-17A to IL-17RA results in tip-to-tip dimerization at a site distal from the ligand-binding site, suggestive of a cytokine-induced priming of the receptors through allostery. The unliganded IL-17RA does not form this interaction, and using SPR, we showed that the IL-25-bound IL-17RB has higher affinity for IL-17RA than does IL-17RB alone (Extended Data Fig. 6a,d,j). From these observations, we conclude that binding of the ligands induces a structural stabilization or rearrangement, perhaps as simple as ordering flexible loops, that results in 'priming' of the tip-to-tip binding site (Extended Data Fig. 8b). Typically, in dimeric ligand–receptor systems such as JAK–STAT cytokines, the cytokine acts as a bridge between the dimers and a single cytokine forms direct contacts with both receptor subunits to act as a crosslinker. In the case of IL-17A and IL-17F, they appear to have the capacity to form both putative non-signalling homodimeric complexes (for example, with IL-17RC) and heterodimeric signalling complexes (for example, IL-17RA–IL-17RC). The equilibrium between putative signalling incompetent binary complexes and signalling competent ternary complexes, in the case of both IL-25 and IL-17A/F, is probably determined by mass action as a function of ligand concentration and receptor expression level on the cell surface, highlighting the complexity of the receptor biology of the IL-17 family.

The signalling-competent heteromeric IL-17–IL-17R complexes adopt a geometry in which the IL-17RA transmembrane domains are spaced far apart from one another, whereas the putative non-signalling IL-25–IL-17RB and IL-17F–IL-17RC homodimeric receptor complexes have transmembrane domains that are close together (Fig. 5h). The spacing of the liganded IL-17RA in our structures is consistent with previous findings showing that the IL-17RA spacing increases upon ligand binding, perhaps to accommodate the size of intracellular adaptor proteins[31]. We hypothesize that the tip-to-tip architecture provides the optimal arrangement of IL-17R intracellular domains required for homotypic IL-17R-SEFIR–ACT1 interactions and signal transduction (Fig. 5h). Further studies are required to determine how the tip-to-tip architecture tunes the spacing and geometry of IL-17–IL-17R complexes and ACT1 signalling.

The use of the anti-IL-17A monoclonal antibody antagonists secukinumab and ixekizumab, as well as the anti-IL-17RA monoclonal antibody brodalumab, which blocks most IL-17 family cytokines, has been successful in the treatment of severe plaque psoriasis[32]. The structural features of the IL-17R tip-to-tip interactions are compatible with

small-molecule targeting, for example, to inhibit aberrant IL-25 signalling in asthma or psoriasis. The structural and organizing principles of IL-17 family cytokine signalling provides a blueprint for the development of both antagonist and agonist therapeutics.

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

## Methods

### Protein expression and purification

For structural studies, proteins were expressed in Expi293 GnTI$^-$ cells (Thermo Fisher). For binding studies and functional studies, proteins were expressed in Expi293F cells (Thermo Fisher). All proteins were expressed at 37 °C in a humidified environment containing 5% $CO_2$. Cell culture supernatants were harvested 3–4 days post-transfection and individual proteins and complexes were purified from the supernatants using Ni-NTA affinity chromatography in 20 mM HEPES pH 7.4 with 150 mM NaCl (HBS 20/150 pH 7.4) with 50 mM imidazole (for washing) and 200 mM imidazole (for elution). Proteins were then further purified by size-exclusion chromatography using a Superdex 200 Increase 10/300 GL column (Cytiva) in HBS 20/150 pH 7.4.

This study used human cytokine and receptor sequences. IL-25 amino acids 30 to 177 (30–177) was expressed with an Igκ signal peptide and C-terminal HRV3C-protease-cleavable(3C) Avitag-6×His tags in the pEG BacMam vector[33] or with a haemagglutinin (HA) signal peptide and C-terminal Avitag-6×His tags under control of the EF-1α promoter and Woodchuck hepatitis virus post-transcriptional response element (WPRE) in a lentiviral vector. IL-17A (24–155) was expressed with an HA signal peptide and C-terminal 3C-Avitag-6×His tags in the pD649 vector (ATUM). IL-17RB (1–288) was expressed with C-terminal 3C-proteinC-8×His tags in the pVLAD6 vector[34] or with C-terminal Avitag-6×His tags in the pD649 vector. IL-17RA (1–317) was expressed with C-terminal 3C-proteinC-8×His tags in the pVLAD6 vector or with C-terminal 3C-proteinC-8×His tags under control of the EF-1α promoter and WPRE in a lentiviral vector. IL-17RC (1–468) (isoform 2) (Uniprot identifier Q8NAC-2) was expressed with C-terminal 3C-proteinC-8×His tags in the pVLAD6 vector.

For biotinylation, proteins with Avitags were site-specifically biotinylated using the biotin ligase BirA and purified using size-exclusion chromatography.

### SPR

Binding studies were conducted on a Biacore T100 (Cytiva). SPR data were collected using Biacore T100 control software version 2.0.4. Approximately, 300 response units (RU) of site-specifically biotinylated IL-17RA, IL-17RB or unrelated negative-control reference protein were immobilized onto a Series S Sensor Chip SA (Cytiva). For each binding cycle, a regeneration condition of 2 M $MgCl_2$ was used. Data were processed using Biacore T100 Evaluation Software, version 2.0.4 (Cytiva). Dissociation constants were calculated using steady-state affinity analysis.

### Cell lines

Cell line sources are as follows: HeLa cells for single-molecule imaging were from the German Collection of Microorganisms and Cell Cultures GmbH (ACC 57); Expi293F cells (Thermo Fisher); Expi293F GnTI$^-$ (Thermo Fisher); IL-17C HEK-Blue NF-κB Reporter cells (Invivogen). Cell lines were authenticated as follows: HeLa cells (German Collection of Microorganisms and Cell Cultures GmbH), Expi293F cells (Thermo Fisher), Expi293F GnTI$^-$ (Thermo Fisher) and IL-17C HEK-Blue NF-κB reporter cells (Invivogen) were guaranteed by the suppliers and no additional authentication was performed by the authors of this study. HeLa cells for single-molecule imaging tested negatively for mycoplasma (PCR). Expi293F cells (Thermo Fisher) and Expi293F GnTI$^-$ cells (Thermo Fisher) used for protein production were not tested for mycoplasma contamination by the authors of this study. The IL-17C HEK-Blue NF-κB reporter cells (Invivogen) were tested for mycoplasma contamination by the manufacturer and no additional testing was performed by the authors of this study.

### NF-κB transcriptional reporter assay

IL-17C HEK-Blue reporter cells (Invivogen) stably expressing human IL-17RA, ACT1, an NF-κB-inducible secreted embryonic alkaline phosphatase (SEAP) reporter gene and endogenous IL-17RB were used to determine NF-κB activity stimulated by IL-25 or IL-17RB mutants. Surface expression of exogenous IL-17RB variants was quantified via epitope staining of the extracellular HA tag and showed overlapping but small differences in expression. The double mutants display greatly reduced signalling efficacy, which was not explained by differences in surface expression. For IL-25 assays, 25,000 cells were seeded overnight into 96-well flat-bottom microplates in the presence of varying doses of IL-25 (WT or mutant). For IL-17RB assays, cells were transfected with plasmid encoding WT or mutant IL-17RB, then seeded into 96-well plates in the presence of WT IL-25 24 h post-transfection. After overnight stimulation, SEAP activity was determined using Quanti-BLUE reagent according to the manufacturer's directions (Invivogen) and read at an absorbance of 640 nm on a SpectraMax Paradigm plate reader using SoftMax Pro v7.1 (Molecular Devices). Cells were maintained according to manufacturer's directions.

### Generation of *IL17R*-knockout cell lines

IL-17C HEK-Blue reporter cells (Invivogen) were transfected in 12-well plates with single guide RNA targeting either *IL17RA* or *IL17RB* and Cas9 nuclease (Synthego) using Lipofectamine CRISPRMAX Cas9 Transfection Reagent (Thermo Fisher). Five days post-transfection, cells were seeded into 10-cm dishes under limiting dilution. After 2–3 weeks, well-separated single colonies were transferred (via cloning discs) into 24-well plates, then expanded and screened for loss of receptor expression and responsiveness to IL-25.

### Human specimen collection

Peripheral blood mononuclear cells (PBMCs) were obtained from the Stanford Blood Center from healthy donors. Written informed consent was obtained from the donors before tissue collection and ethics oversight was ensured by the Stanford Blood Center.

### Bulk PBMC IL-5 secretion assay

PBMCs from healthy donors were obtained from the Stanford Blood Center. Human PBMCs were isolated from donor LRS chambers using Ficoll density gradient centrifugation (Cytiva) and cryopreserved until time of use. The assay was adapted from ref. [35] with the following modifications: PBMCs were stimulated in 96-well round-bottom plates with varying concentration of IL-25 (WT or mutants) in the presence of 1 nM MSA-hIL-2 and 0.7 nM (approximately 10 ng ml$^{-1}$) rhTSLP (Peprotech). After 72 h, supernatants were collected and the concentration of IL-5 was determined by ELISA (R&D systems) read at an absorbance of 450 nm on a SpectraMax Paradigm plate reader using SoftMax Pro v7.1 (Molecular Devices).

### Single-molecule imaging and analysis

For cell surface labelling, IL-17 receptors were N-terminally fused to suitable tags using a pSems vector including the signal sequence of Igκ (pSems-leader)[36]. IL-17RA (24–866) was fused to non-fluorescent monomeric GFP (mXFP)[37]; IL-17RB (18–502) was fused to the ALFA-tag[38]. HeLa cells (ACC 57, DSMZ Germany) were cultured and transiently transfected as previously described[36]. Anti-GFP and anti-ALFAtag nanobodies site specifically labelled with ATTO 488, Cy3b, Rho11 or ATTO 643 via a C-terminal cysteine[30] were added at concentrations of 2 nM and 2.5 nM, respectively, at least 10 min before imaging.

Single-molecule imaging was carried out by dual-colour total internal reflection fluorescence microscopy using an inverted microscope (IX71, Olympus) equipped with a spectral image splitter (DualView, Optical Insight) and a back-illuminated electron multiplied CCD camera (iXon DU897D, Andor Technology). Fluorophores were excited by simultaneous illumination with a 561-nm laser (CrystaLaser; approximately 32 W cm$^{-2}$) and a 642-nm laser (Omicron: approximately 22 W cm$^{-2}$). Image stacks of 150 frames were recorded for each cell at a time resolution of 32 ms per frame, with at least 10 cells recorded in each

experiment. Ligands were incubated for 15 min before imaging. All imaging experiments were carried out at room temperature.

Dual-colour time-lapse images were evaluated using an in-house developed MATLAB software (SLIMfast4C, https://zenodo.org/record/5712332) as previously described in detail[30]. After channel registration based on calibration with fiducial markers, molecules were localized using the multi-target tracking algorithm[39]. Immobile emitters were filtered out by spatiotemporal cluster analysis[40]. For co-tracking, frame-by-frame co-localization within a cut-off radius of 100 nm was applied followed by tracking of co-localized emitters using utrack[41]. Molecules co-diffusing for 10 frames or more were identified as dimers. Relative levels of homodimerization and heterodimerization were determined based on the fraction of co-localized particles[30]. Diffusion properties were determined from pooled single trajectory using mean squared displacement analysis for all trajectories with a lifetime greater than 10 frames. Diffusion constants were determined from the mean squared displacement by linear regression.

## Cryo-EM
Aliquots of 3 µl of IL-25–IL-17RB, IL-25–IL-17RB–IL17RA or IL-17RA–IL-17A–IL-17RC complexes were applied to glow-discharged Quantafoil or AuUltrafoil (1.2/1.3) grids. The IL-25–IL-17RB, IL-25–IL-17RB–IL-17RA grids were blotted for 1–2 s at 100% humidity with an offset of −15 and plunge frozen into liquid ethane using a Vitrobot Mark IV (Thermo Fisher). For the IL-17RA–IL-17A–IL-17RC complexes, 0.01% fluorinated octyl maltoside (Anatrace) was added before vitrification; the grids were blotted for 3.5 s at 100% humidity with an offset of −1 and plunge frozen into liquid ethane using a Vitrobot Mark IV (Thermo Fisher). IL-25–IL-17RB and IL-25–IL-17RB–IL-17RA grids were imaged on a 300 keV Titan Krios cryo-electron microscope (Thermo Fisher) equipped with a K3 camera (Gatan) at the Stanford Cryo-EM Center (cEMc) and IL-17RA–IL-17A–IL-17RC grids were imaged on a 300 keV Titan Krios cryo-electron microscope (Thermo Fisher) equipped with a K3 camera (Gatan) at the Howard Hughes Medical Institute (HHMI) Janelia Research Campus Cryo-EM Facility. Videos were collected at a calibrated magnification corresponding to a 0.8521 Å per physical pixel at the cEMc and 1.06 Å per physical pixel at Janelia. The dose was set to a total of 50–53 electrons per Å². Automated data collection was carried out using SerialEM with a nominal defocus range set in increments between −0.8 and −2.2 µM. All grids were imaged at a stage tilt of 0°. IL-25–IL-17RB grids were additionally imaged at a stage tilt of 40° and IL-25–IL-17RB–IL-17RA grids were additionally imaged at a stage tilt of 35°.

## Cryo-EM image processing
Videos were processed using cryoSPARC v3.1.0 or v3.2.0 (ref. [42]). Videos were motion corrected, had contrast transfer functions determined and particles picked using the cryoSPARC live processing functions. During this processing, micrographs were binned to the physical pixel size.

## Cryo-EM data processing
All 2D classifications and 3D reconstructions were performed using cryoSPARC v3.1.0 or v3.2.0. For the IL-25–IL-17RB 2:2 binary complex, particles were kept apart by their tilt status and reference-free 2D classification was performed, leaving 1,125,580 particles in well-defined classes. From here, a subset of 51,000 0° tilt particles and 83,439 40° tilt particles were selected to generate an ab initio model. This ab initio model was then used in two rounds of iterative heterogenous refinement against a junk class, using the 1,125,580-particle stack. This resulted in a class with 712,571 particles, which had a resolution of 3.53 Å when refined with non-uniform refinement[43]. A generous mask around the 2:2 complex was used in local refinement with a fulcrum at centre mass, which resulted in a 3.43 Å reconstruction. The particles were then motion corrected again using per-particle motion correction. After the motion correction, the local refinement was run again, resulting

in a 3.18 Å reconstruction. This map was then sharpened using DeepEMhancer[44] on the COSMIC[2,45] webserver.

For the IL-25–IL-17RB 6:6 binary complex, the initial processing was as in the 2:2 complex to generate the initial 2D averages. At this point, the untilted and tilted particles were combined and re-extracted at a 2.3 times bigger box size. Two-dimensional classification was run and classes containing obvious 6:6 complex were selected, leaving 297,300 particles. The particles were selected to generate three ab initio models, which were then processed with one round of heterogenous refinement. This resulted in a class with 131,696 particles, which had a resolution of 4.39 Å when refined with non-uniform refinement[43]. This map was then sharpened using DeepEMhancer[44] on the COSMIC[2,45] webserver.

For the IL-25–IL-17RB–IL-17RA 2:2:2 ternary complex, particles were kept apart by their tilt status, and reference-free 2D classification was performed and a subset of 42,208 0° tilt particles and 55,376 35° tilt particles were selected to generate an ab initio model using C2 symmetry. This ab initio model was then used in four rounds of iterative heterogenous refinement against a junk class, using a 1,287,497-particle stack. This resulted in a class with 509,311 particles, which had a resolution of 3.86 Å when refined with non-uniform refinement[43]. The particles were then motion corrected again using per-particle motion correction. After the motion correction, the non-uniform refinement was run again, resulting in a 3.66 Å reconstruction. This map was then sharpened using DeepEMhancer[44] on the COSMIC[2,45] webserver.

For the IL-17-RA–IL-17RA 2:2 binary complex, reference-free 2D classification was performed and a subset of 323,298 0° tilt particles were selected to generate an ab initio model. This model was then used in four rounds of iterative heterogenous refinement against a junk class and a class representing the ternary complex, using 6,529,311 particles. This resulted in a class with 2,986,310 particles, which had a resolution of 2.51 Å when refined with non-uniform refinement[43]. This map was then sharpened using DeepEMhancer[44].

For the IL-17-RA–IL-17A–IL-17RC 2:2:1 ternary complex, reference-free 2D classification was performed and a subset of 98,949 0° tilt particles representing distinct ternary views were selected to generate an ab initio model. This model was then used in six rounds of iterative heterogenous refinement against a junk class and a class representing the binary complex, using 6,529,311 particles. This resulted in a class with 542,344 particles, which had a resolution of 3.01 Å when refined with non-uniform refinement[43]. This ternary map was seen to have more orientational bias and anisotropy in comparison to the binary complex, probably due to issues sorting between binary and ternary classes at certain angles. This map was then sharpened using DeepEMhancer[44].

## Model refinement and structural analysis
In general, models of crystal structures, cryo-EM structures or Alphafold[46] models were manually docked into cryo-EM maps in UCSF Chimera or UCSF ChimeraX[47]. Models were then refined using rigid-body refinement with Phenix[48] followed by refinement with ISOLDE[49], and further iterative manual building and refinement in Coot[50] and Phenix.

For the higher-order binary complex, refined IL-25–IL-17RB binary complex models were docked into the higher-order map and refined using real-space refinement in Phenix. Regions of the higher-order model that did not have corresponding density were removed and glycans were not modelled.

For the IL-25–IL-17RB–IL-17RA ternary complex, the IL-25–IL-17RB binary complex and the IL-17RA crystal structure from Protein Data Bank (PDB) ID 3JVF were docked into the map and refined manually and using real-space refinement in Phenix. Owing to the quality of the density for the D2 domains of IL-17RA, the D2 domain of 3JVF was docked into the map and did not undergo manual refinement.

For the IL-17–IL-17RA–IL-17RC ternary complex, the IL-17–IL-17RA binary complex and the IL-17RA crystal structure from PDB ID 6HG9 were docked into the map and refined manually and using real-space refinement in Phenix. Owing to the quality of the electron density for

the D3 and D4 domains of IL-17RC, the D3 and D4 domains of 6HG9 were docked into the map and did not undergo manual refinement.

Interfaces were analysed using PISA, ChimeraX and PyMOL[51](Schrödinger). All structural figures were made using ChimeraX.

## Statistics

Sample sizes were not pre-determined using statistical methods. Investigators were not blinded to the experimental sample identities. Single-molecule data were tested for normality, and owing to non-normality of some of the samples, non-parametric two-sample Kolmogorov–Smirnov tests were used to calculate differences between samples. Data were plotted and statistics were calculated using Prism 9 (GraphPad). Standard box and whisker plots used for the single-molecule imaging data show the five number summaries of the data: minimum, first quartile, median, third quartile and maximum values. $P$ values are indicated in the figures as: $*P < 0.05$, $**P < 0.01$, $***P < 0.001$ and $****P < 0.0001$. Cryo-EM data and refinement statistics are provided in Extended Data Table 1.

## Reporting summary

Further information on research design is available in the Nature Research Reporting Summary linked to this article.

## Data availability

Cryo-EM maps and atomic coordinates for the IL-17RB–IL-25 (2:2), IL-17RB–IL-25 (6:6), IL-17RB–IL-25–IL-17RA (2:2:2), IL-17RA–IL-17A (2:2) and IL-17RA–IL-17A–IL-17RC (2:2:1) structures have been deposited in the Electron Microscopy Data Bank (EMD-26833, EMD-26834, EMD-26835, EMD-26836 and EMD-26837) and the PDB (7UWJ, 7UWK, 7UWL, 7UWM and 7UWN), respectively. All single-molecule imaging data have been deposited at Zenodo: raw images are at https://doi.org/10.5281/zenodo.6783369 and calibration images are at https://doi.org/10.5281/zenodo.6787325. Source data are provided with this paper. Other data and materials are available from the corresponding author on reasonable request. Source data are provided with this paper.

## Code availability

The in-house developed software for localization-based imaging in MATLAB (SLIMfast4C) is available at https://zenodo.org/record/5712332.

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

**Acknowledgements** We thank L. Montabana and C. Zhang from the Stanford University Cryo-Electron Microscopy Center (cEMc), and R. Yan and S. Yang from the Howard Hughes Medical Institute Janelia Research Campus Cryo-EM Facility for their assistance with data collection. Structural biology applications used in this project were compiled and configured by SBGrid[52]. We thank members of the Garcia laboratory for support and helpful discussions; R. Saxton for critical review of the manuscript; and Y. Miao and D. Siepe for advice on the SPR experiments. N.A.C. is supported by a CIHR postdoctoral fellowship. M.Y. is supported by the Stanford Immunology Program training grant (5T32AI07290-33) and the Stanford Immunology and Rheumatology training grant (5T32AR050942-15). J.P. was supported by the DFG (Imaging Facility iBiOs, PI 405/14-1 and SFB 944/P8, INST 190/146-3). K.C.G. is an investigator with the Howard Hughes Medical Institute, and is supported by a Sanofi Innovation Award and US National Institutes of Health grant R01-AI51321.

**Author contributions** K.C.G. conceived of the project. K.C.G. and S.C.W. designed the experimental approach. S.C.W., K.C.G. and N.A.C. wrote the manuscript with input from all authors. S.C.W. and X.X. purified all recombinant proteins. S.C.W., N.A.C. and N.T. collected the cryo-EM data. S.C.W. and N.A.C. processed the cryo-EM data. S.C.W., N.A.C. and K.M.J. refined and analysed the cryo-EM models. M.Y. performed the cell signalling assays. C.P. performed and analysed single-molecule imaging studies with support from M.H, and supervised by J.P. S.C.W. performed the SPR binding studies. K.C.G. and J.P. supervised the research.

**Competing interests** The Garcia laboratory was funded, in part, by Sanofi to carry out the research in this paper.

**Additional information**
**Correspondence and requests for materials** should be addressed to K. Christopher Garcia.

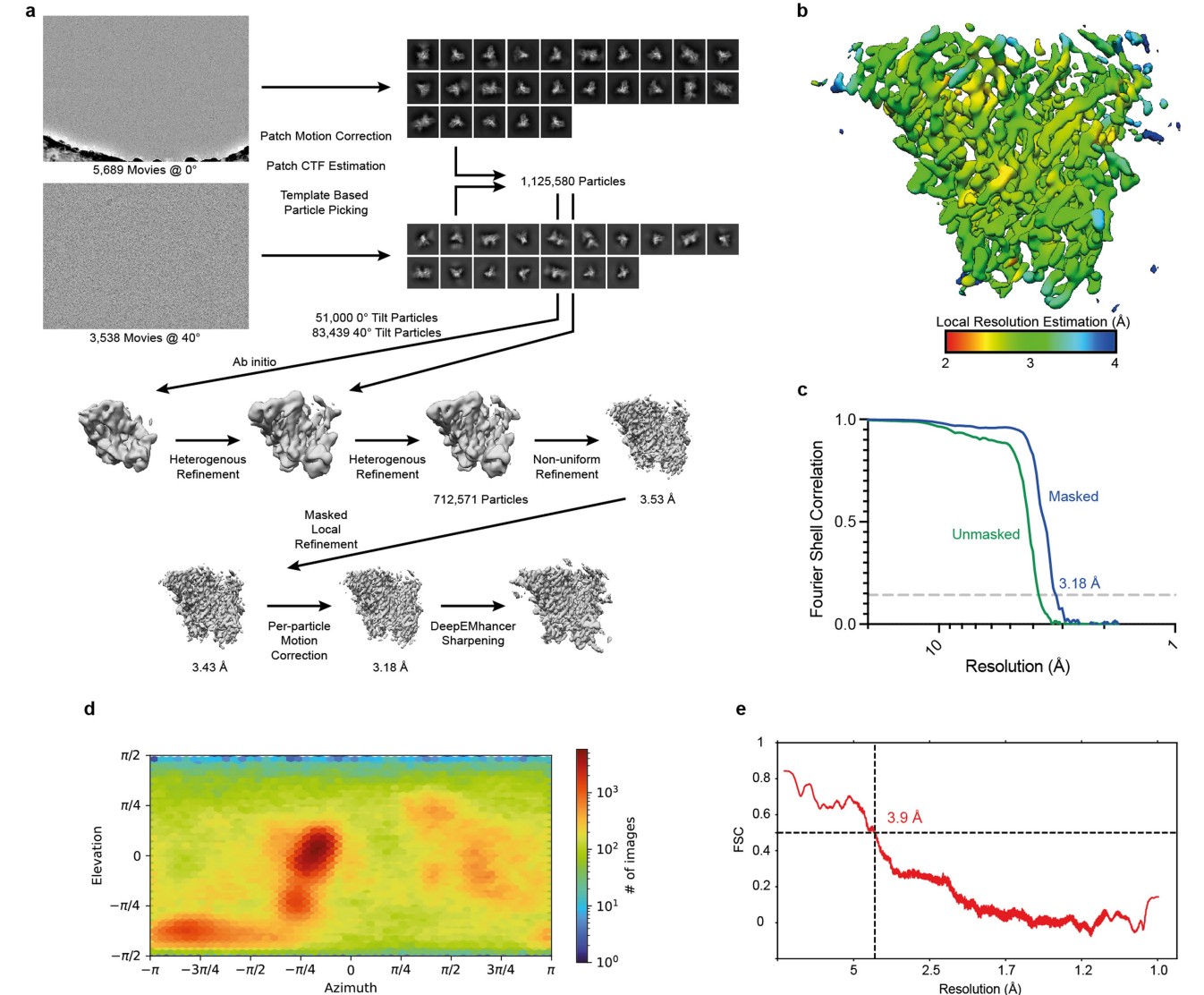

**Extended Data Fig. 1 | IL-25–IL-17RB binary 2:2 complex cryo-EM data processing. a**, Workflow for cryo-EM data processing. Representative micrographs, reference free 2D averages, and cryo-EM maps at the various stages of processing. **b**, Local resolution estimation of the finalized cryo-EM map[42]. **c**, FSC curve of the complex reconstruction using gold-standard refinement calculated from unmasked and masked half maps. **d**, Orientational distribution of the complex reconstruction. **e**, Map vs model FSC for the 2:2 IL-25–IL-17RB binary complex.

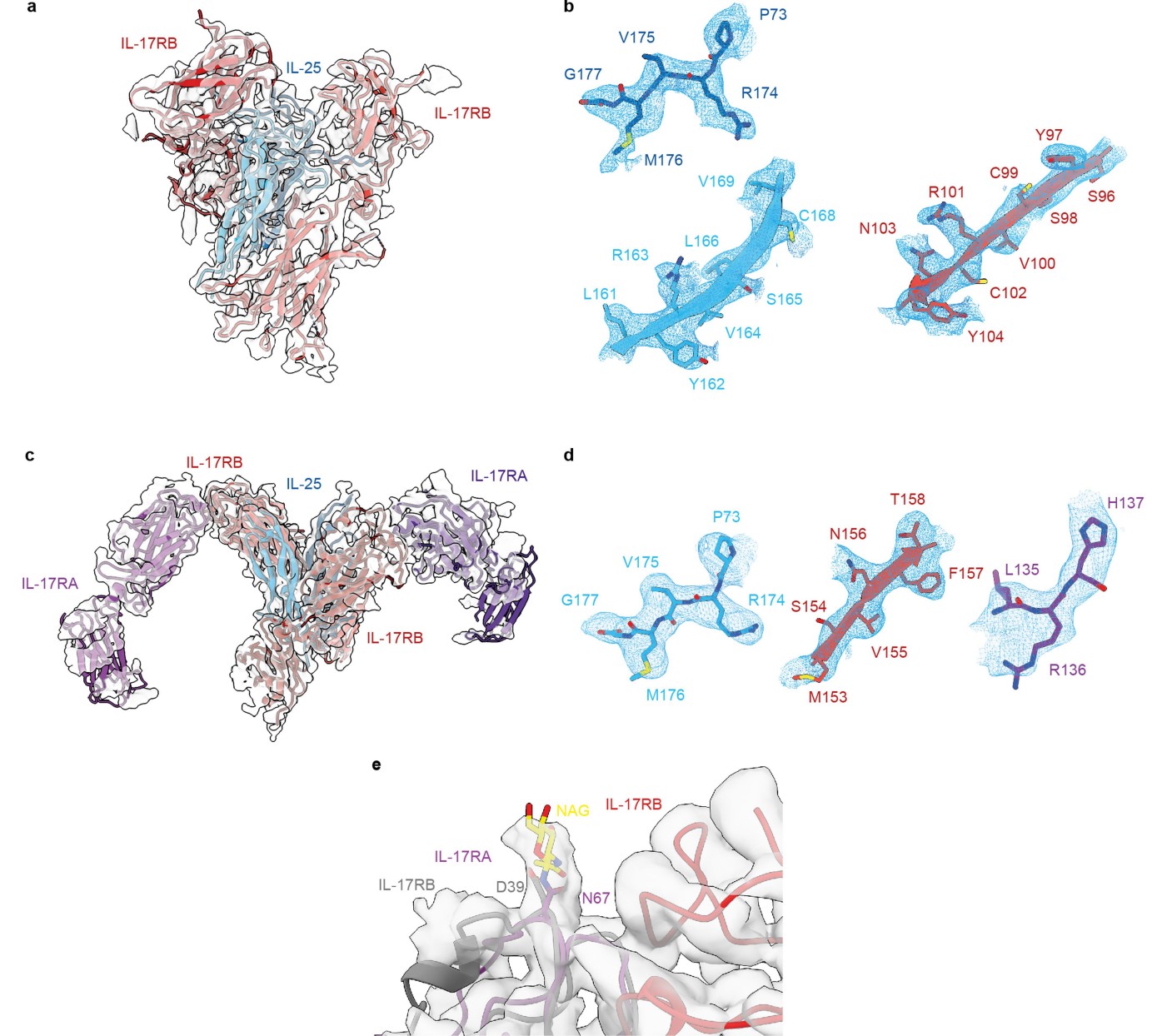

**Extended Data Fig. 2 | Cryo-EM density of IL-25–IL-17RB binary and IL-25–IL-17RB–IL-17RA ternary complexes. a**, Cryo-EM density of the IL-25 binary 2:2 complex with the model fitted. IL-25 is in blue and IL-17RB is in red. **b**, Selected regions of the map and model as in **a**. **c**, Cryo-EM density of the IL-25 ternary 2:2:2 complex with the model fitted. IL-25 is in blue, IL-17RB is in red, and IL-17RA is in purple. **d**, Selected regions of the map and model as in **c**. **e**, A resolved IL-17RA glycan, N67-NAG, near the tip-to-tip interface distinguishes IL-17RA from IL-17RB in the wing of the ternary complex in **c**. Models of IL-17RB (red and grey) and IL-17RA (purple) are docked into the cryo-EM density of the IL-25 ternary complex map.

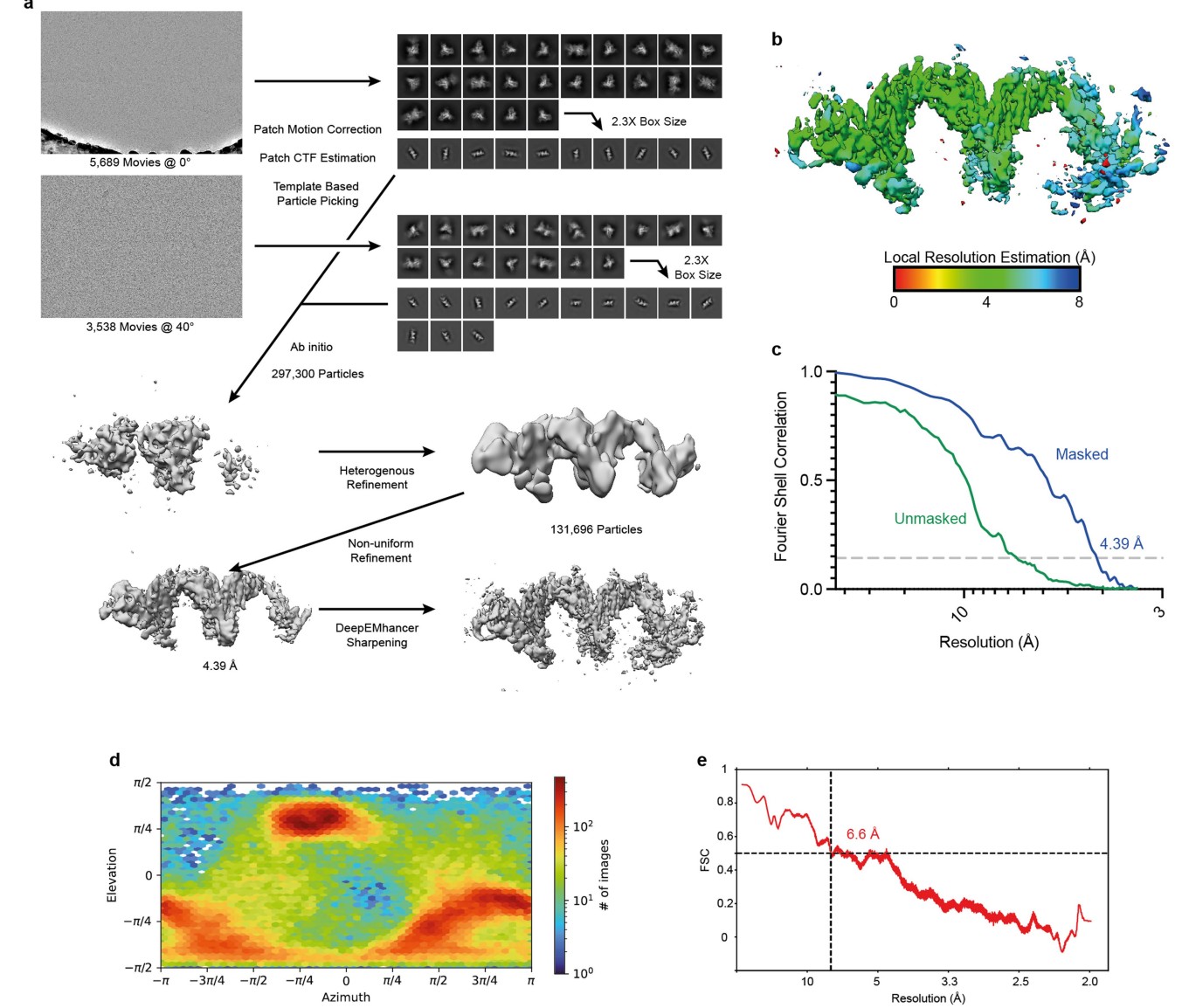

**Extended Data Fig. 3 | IL-25–IL-17RB binary 6:6 complex cryo-EM data processing. a**, Workflow for cryo-EM data processing. Representative micrographs, reference free 2D averages, and cryo-EM maps at the various stages of processing. **b**, Local resolution estimation of the finalized cryo-EM map[42]. **c**, FSC curve of the complex reconstruction using gold-standard refinement calculated from unmasked and masked half maps. **d**, Orientational distribution of the complex reconstruction. **e**, Map vs model FSC for the 6:6 IL-25–IL-17RB binary complex.

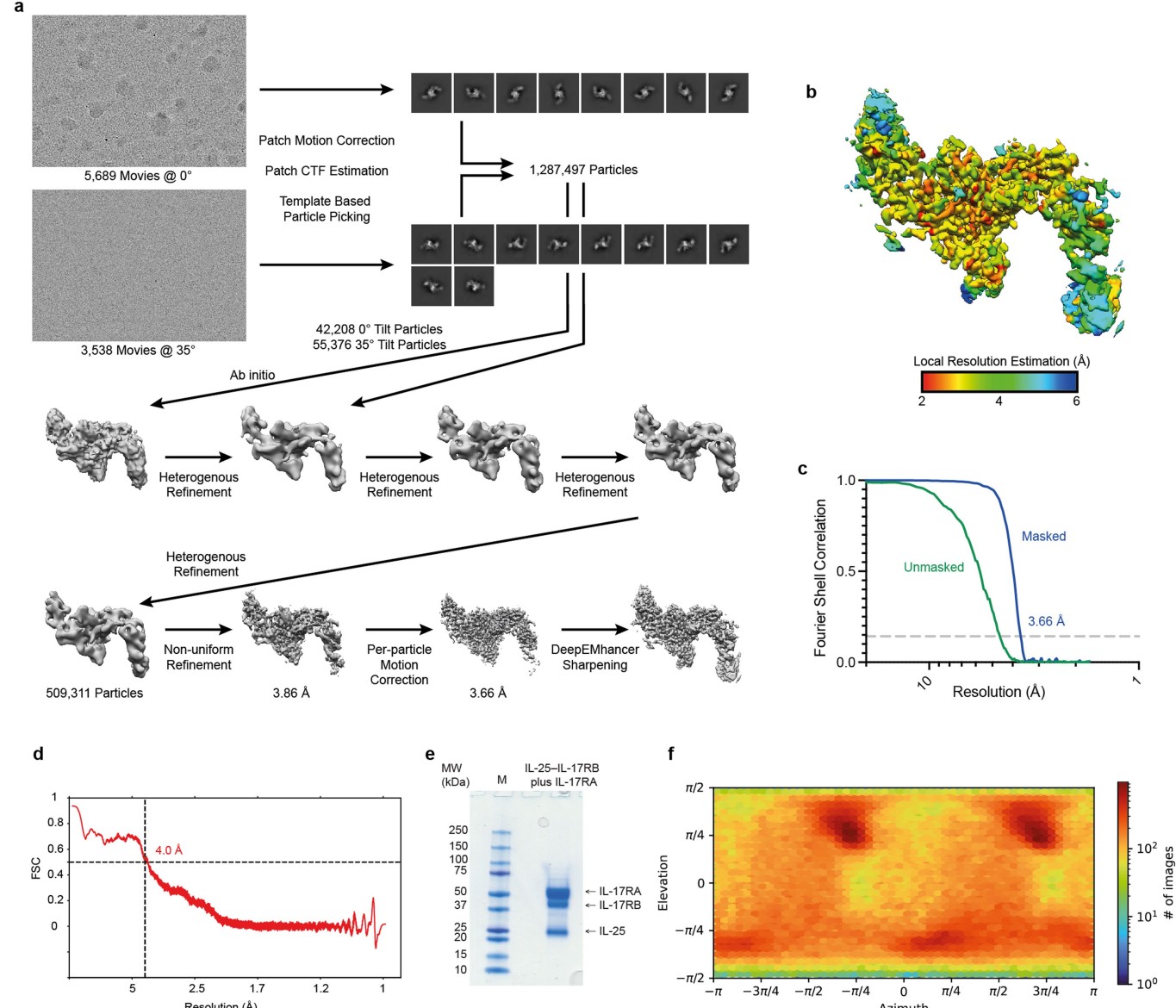

**Extended Data Fig. 4 | IL-25–IL-17RB–IL-17RA ternary 2:2:2 complex cryo-EM data processing. a**, Workflow for cryo-EM data processing. Representative micrographs, reference free 2D averages, and cryo-EM maps at the various stages of processing. **b**, Local resolution estimation of the finalized cryo-EM map[42]. **c**, FSC curve of the complex reconstruction using gold-standard refinement calculated from unmasked and masked half maps. **d**, Map vs model FSC for the 2:2:2 IL-25–IL-17RB–IL-17RA ternary complex. **e**, SDS-PAGE gel image showing the sample of purified IL-17RA mixed with IL-25–IL-17RB that was imaged using cryo-EM. "M" = molecular weight marker. **f**, Orientational distribution of the complex reconstruction.

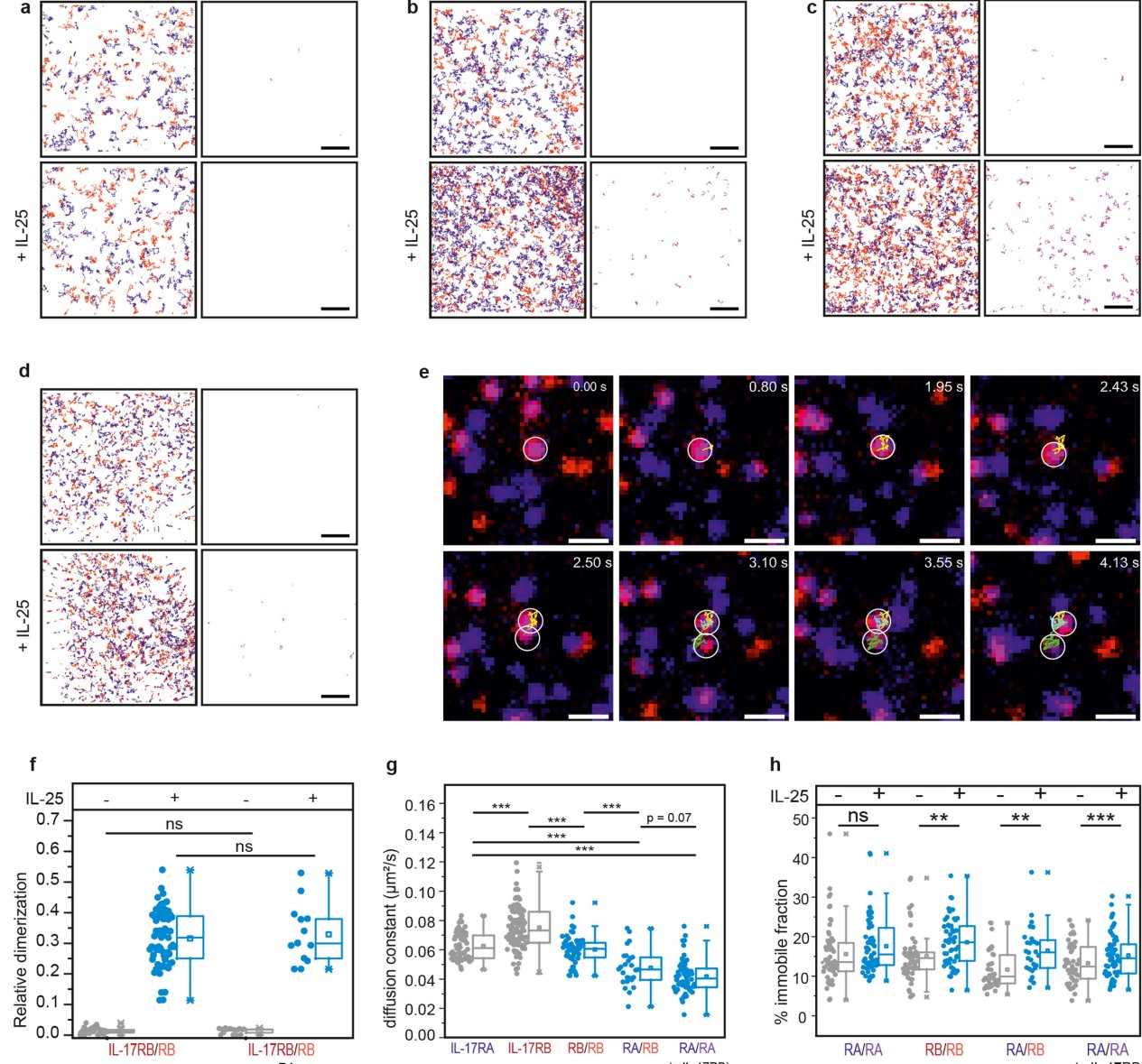

**Extended Data Fig. 5 | Supplementary single-molecule imaging data.**
**a**–**d**, Single-molecule trajectories (red, blue, left) and co-trajectories (magenta, right) before (top) and after stimulation with IL-25 shown for representative experiments: **a**, Homodimerization of IL-17RA (red, blue); **b**, heterodimerization of IL-17RA (red) and IL-17RB (blue); **c**, homodimerization of IL-17RB (red, blue); **d**, homodimerization of IL-17RA (red, blue) in the presence of IL-17RB. Scale bar: 5 μm in all images. **e**, Time-lapse images following a representative dissociation of an IL-17RB oligomer into two individual IL-17RB homodimers. Scale bar: 2 μm. **f**, control experiment comparing IL-17RB homodimerization in the absence and presence of IL-17RA.

Numbers of cells/experiments are 25/4, 49/4, 17/1, and 14/1 for each column, going from left to right. **g**, Diffusion constants of IL-17RA and IL-17RB for individual experiments. Numbers of cells/experiments are 60/2, 100/4, 54/4, 26/3, and 62/6 for each column, going from left to right. **h**, Percentage of immobilized fractions for individual experiments. Numbers of cells/experiments are 60/2, 62/2, 100/4, 108/4, 62/3, 56/3, 162/6, and 180/6 for each column, going from left to right. In **f**–**h**, each data point represents the result from one cell. Statistical differences were calculated with two-sample Kolmogorov-Smirnov tests (**P ≤ 0.01, ***P ≤ 0.001; ns: not significant).

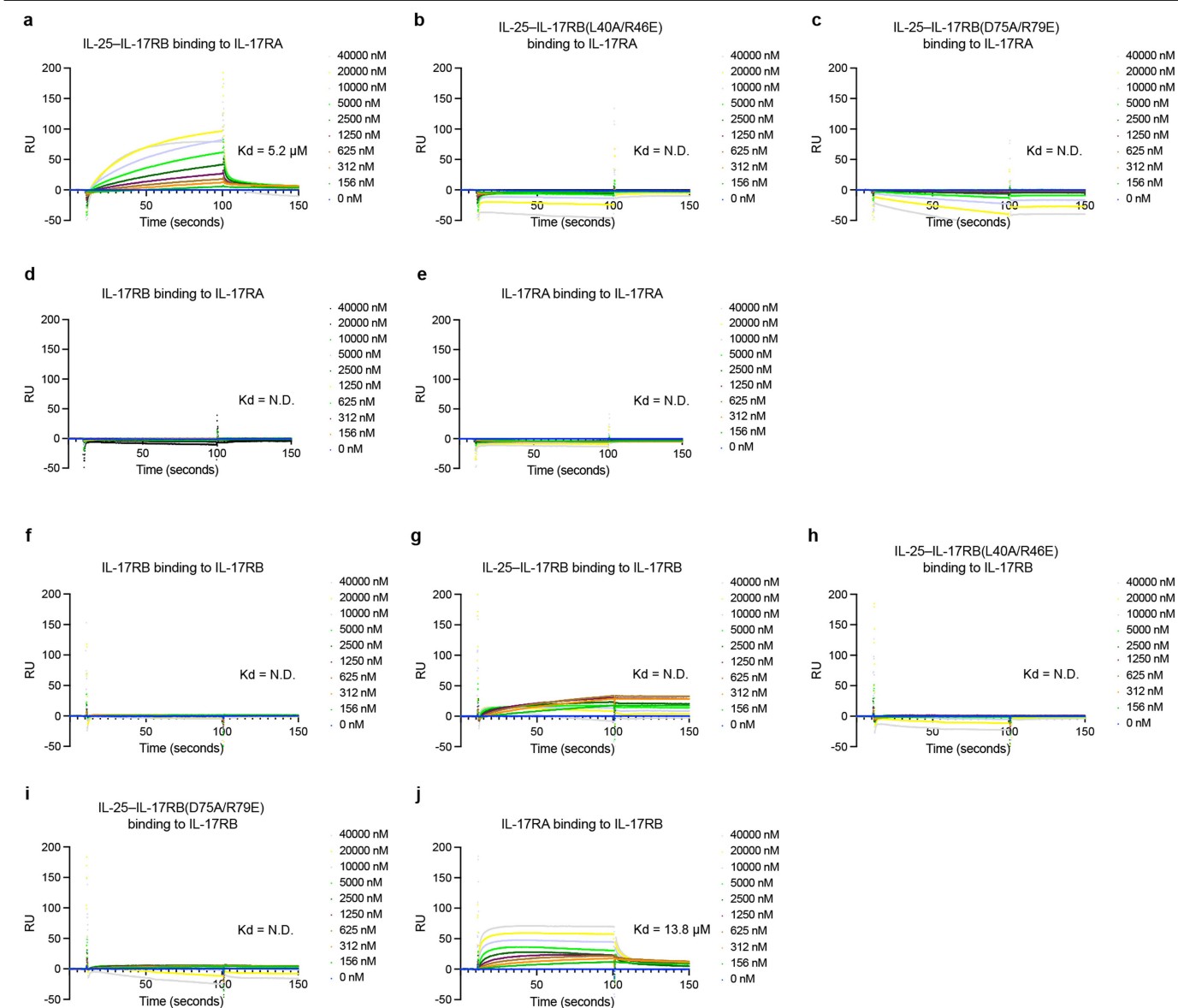

**Extended Data Fig. 6 | Supporting surface plasmon resonance (SPR) binding data for IL-25, IL-17RB, and IL-17RA interactions. a**, sensorgram showing wt IL-25–IL-17RB complex binding to immobilized IL-17RA. **b**, sensorgram showing no binding of IL-25–IL-17RB(L40A/R46E) mutant complex to immobilized IL-17RA. **c**, sensorgram showing no binding of IL-25–IL-17RB(D75A/R79E) mutant complex to immobilized IL-17RA. **d**, sensorgram showing no binding of IL-17RB to immobilized IL-17RA. **e**, sensorgram showing no binding of IL-17RA to immobilized IL-17RA. **f**, sensorgram showing no binding of IL-17RB to immobilized IL-17RB. **g**, sensorgram showing wt IL-25–IL-17RB complex binding to immobilized IL-17RB. **h**, sensorgram showing no binding of IL-25–IL-17RB(L40A/R46E) mutant complex to immobilized IL-17RB. **i**, sensorgram showing no binding of IL-25–IL-17RB(D75A/R79E) mutant complex to immobilized IL-17RB. **j**, sensorgram showing weak binding of IL-17RA to immobilized IL-17RB. Analyte concentration ranges are shown next to each sensorgram. Dissociation constants (Kd) are indicated on the sensorgrams. N.D. = Not Determined. Single experiments (*n* = 1) were performed for **b**, **c**, and **e**–**j**. Sensorgrams for **a** and **d** are representative of a single experiment from two independent experiments (*n* = 2).

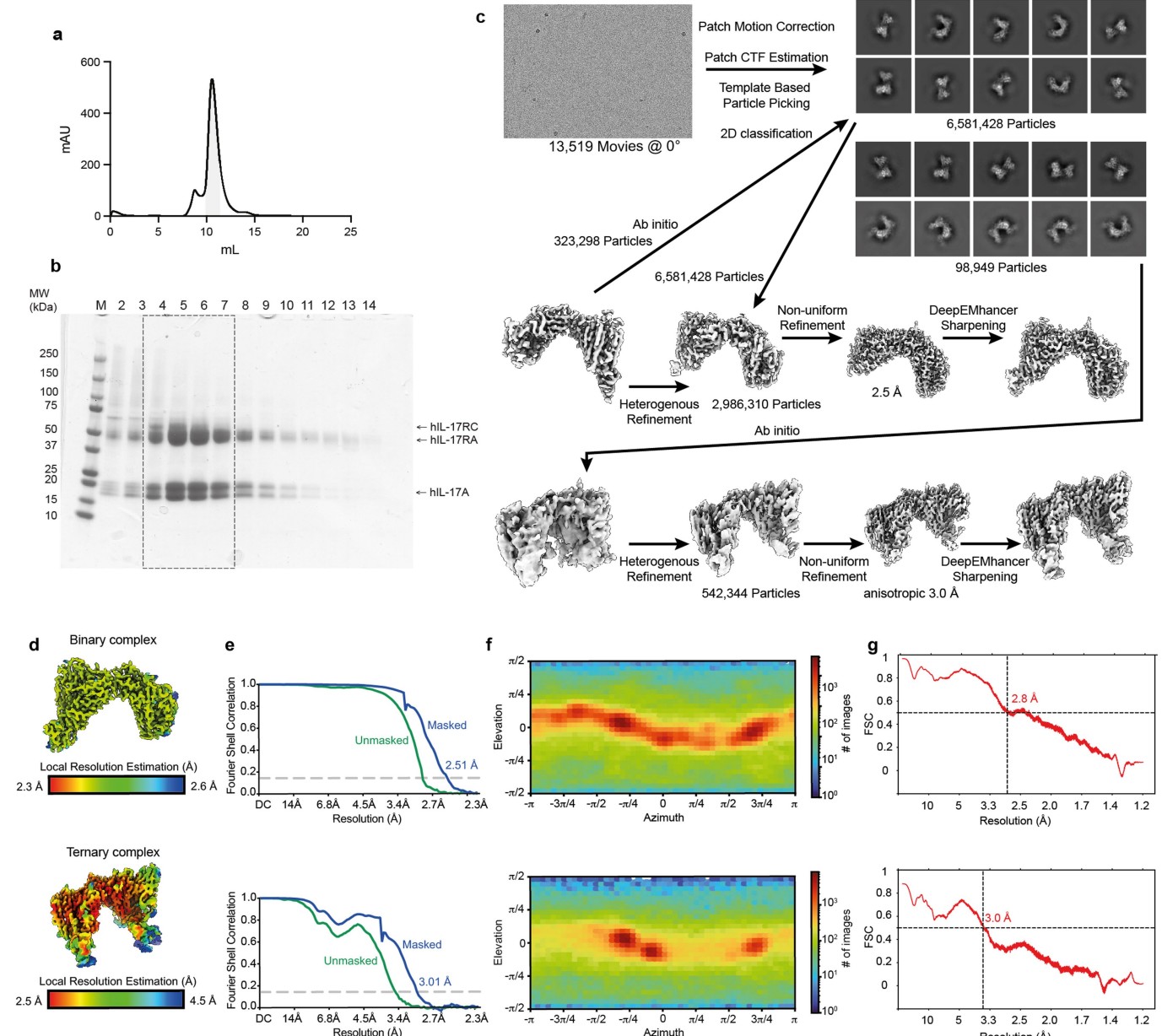

**Extended Data Fig. 7 | IL-17A binary and ternary complex cryo-EM data processing. a**, Superdex 200 size-exclusion chromatography profile of IL-17A binary and ternary complexes. **b**, SDS-PAGE gel of SEC fractions. Grey region under the trace in **a**, is outlined with a dotted box in **b**. These fractions were used in cryo-EM analysis. **c**, Workflow for cryo-EM data processing. Representative micrograph, reference free 2D averages, and cryo-EM maps at the various stages of processing. **d**, Local resolution estimation of the finalized cryo-EM maps[42]. **e**, FSC curves of the complex reconstructions using gold-standard refinement calculated from unmasked and masked half maps. **f**, Orientational distributions of the complex reconstructions. **g**, Map vs model FSCs for the IL-17A binary and ternary complexes.

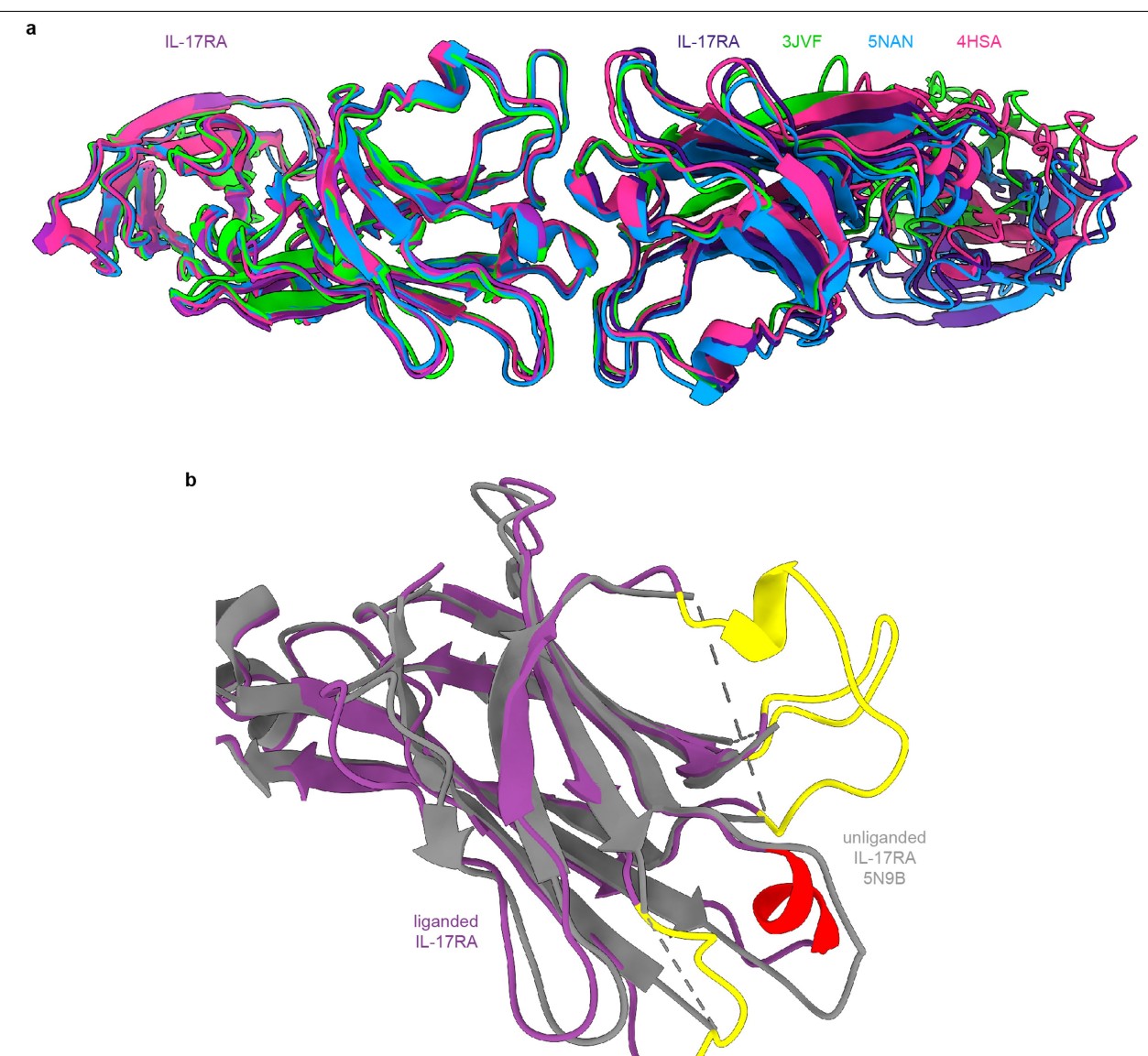

**Extended Data Fig. 8 | IL-17RA structural overlay. a**, IL-17RA from the cryo-EM analysis superimposed with the 3JVF crystal structure, 5NAN crystal structure, and 4HSA crystal structure, in purple, green, blue, and pink, respectively. **b**, IL-17RA is stabilized by IL-17A binding. IL-17RA from the cryo-EM analysis superimposed with the unliganded 5N9B crystal structure, in purple and grey, respectively. Regions where the unliganded structure has unresolved loops are colored in yellow, while regions where the unliganded structure has less ordered secondary structure are colored in red.

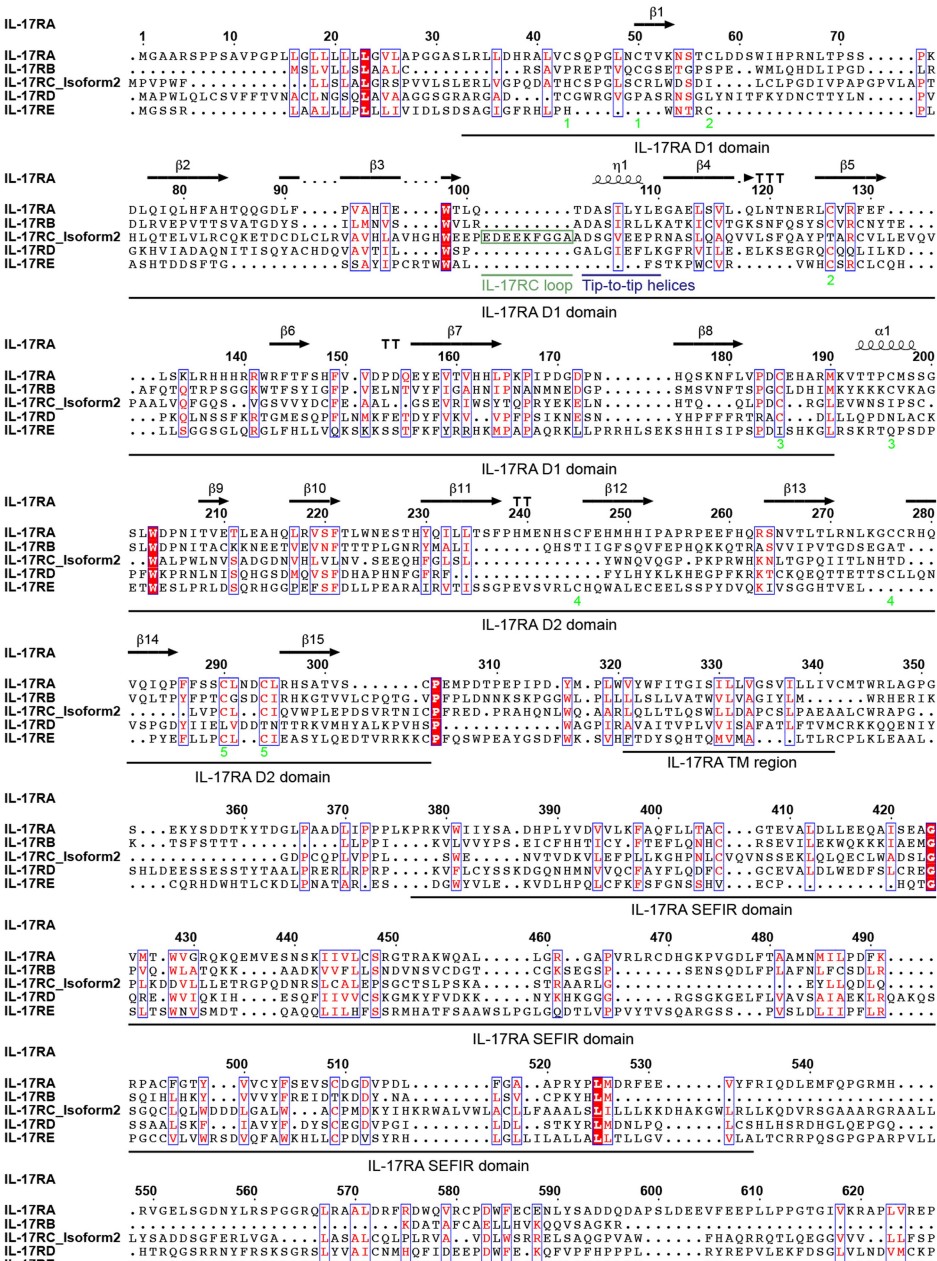

**Extended Data Fig. 9 | Sequence alignment of human IL-17RA–E.** Sequences were aligned using MUSCLE[53] and displayed, along with the secondary structure elements of IL-17RA from the IL-25–IL-17RB–IL-17RA ternary structure, using ESPript 3.0[54]. Disulfide bonds are annotated with bright green letters. The D1 domain, D2 domain, TM region, and SEFIR domain of IL-17RA are underlined in black. Note the conservation of the region near the tip-to-tip interfaces as well as the insertion of the negatively charged loop in IL-17RC.

Extended Data Table 1 | Cryo-EM data collection, refinement and validation statistics

| | Binary IL-25–IL-17RB complex (EMDB-26833) (PDB 7UWJ) | Higher-order binary IL-25–IL-17RB complex (EMDB-26834) (PDB 7UWK) | Ternary IL-25–IL-17RB–IL-17RA complex (EMDB-26835) (PDB 7UWL) | Binary IL-17–IL-17RA complex (EMDB-26836) (PDB 7UWM) | Ternary IL-17RA–IL-17–IL-17RC complex (EMDB-26837) (PDB 7UWN) |
|---|---|---|---|---|---|
| **Data collection and processing** | | | | | |
| Magnification | 29000× | 29000× | 29000× | 81000× | 81000× |
| Voltage (kV) | 300 | 300 | 300 | 300 | 300 |
| Electron exposure (e–/Å$^2$) | 52.5 | 52.5 | 52.5 | 50 | 50 |
| Defocus range (μm) | -0.8 to -2.2 | -0.8 to -2.2 | -0.8 to -2.2 | -0.8 to -2.0 | -0.8 to -2.0 |
| Pixel size (Å) | 0.8521 | 0.8521 | 0.8521 | 1.066 | 1.066 |
| Symmetry imposed | C1 | C1 | C2 | C1 | C1 |
| Initial particle images (no.) | 1,125,580 | 1,125,580 | 1,287,497 | 6,529,311 | 6,529,311 |
| Final particle images (no.) | 712,571 | 297,300 | 509,311 | 2,986,310 | 542,344 |
| Map resolution (Å) | 3.18 | 4.39 | 3.66 | 2.51 | 3.01 |
| FSC threshold | 0.143 | 0.143 | 0.143 | 0.143 | 0.143 |
| | | | | | |
| **Refinement** | | | | | |
| Initial model used (PDB or AlphaFold code) | AF-Q9H293-F1, AF-Q9NRM6-F1 | AF-Q9H293-F1, AF-Q9NRM6-F1 | AF-Q9H293-F1, AF-Q9NRM6-F1, 3JVF | 4HSA, 3JVF | 4HSA, 3JVF |
| Model resolution (Å) | 3.9 | 6.6 | 4.0 | 2.8 | 3.5 |
| FSC threshold | 0.5 | 0.5 | 0.5 | 0.5 | 0.5 |
| Map sharpening B factor (Å$^2$) | -159.5 | -185.6 | -157.3 | -122.2 | -120.8 |
| Model composition | | | | | |
| Non-hydrogen atoms | 5316 | 16070 | 10102 | 7972 | 10602 |
| Protein residues | 664 | 2059 | 1236 | 967 | 1325 |
| Ligands | NAG:6 | N/A | NAG:22 | NAG:9 | NAG:11 |
| B factors (Å$^2$) | | | | | |
| Protein | 85.29 | 175.37 | 124.92 | 45.91 | 94.28 |
| Ligand | 87.51 | N/A | 124.31 | 62.42 | 96.34 |
| R.m.s. deviations | | | | | |
| Bond lengths (Å) | 0.004 | 0.004 | 0.004 | 0.003 | 0.004 |
| Bond angles (°) | 0.777 | 0.801 | 0.746 | 0.595 | 0.622 |
| | | | | | |
| Validation | | | | | |
| MolProbity score | 1.84 | 2.10 | 2.14 | 1.89 | 2.34 |
| Clashscore | 11.24 | 17.97 | 15.36 | 6.68 | 14.46 |
| Poor rotamers (%) | 0 | 0.11 | 0.63 | 2.96 | 2.21 |
| Ramachandran plot | | | | | |
| Favored (%) | 95.94 | 94.96 | 92.93 | 97.04 | 93.88 |
| Allowed (%) | 4.06 | 5.04 | 7.07 | 2.96 | 6.12 |
| Disallowed (%) | 0 | 0 | 0 | 0 | 0 |

# Reporting Summary

## Statistics

For all statistical analyses, confirm that the following items are present in the figure legend, table legend, main text, or Methods section.

| n/a | Confirmed | |
|-----|-----------|---|
| ☐ | ☒ | The exact sample size (*n*) for each experimental group/condition, given as a discrete number and unit of measurement |
| ☐ | ☒ | A statement on whether measurements were taken from distinct samples or whether the same sample was measured repeatedly |
| ☐ | ☒ | The statistical test(s) used AND whether they are one- or two-sided<br>*Only common tests should be described solely by name; describe more complex techniques in the Methods section.* |
| ☒ | ☐ | A description of all covariates tested |
| ☐ | ☒ | A description of any assumptions or corrections, such as tests of normality and adjustment for multiple comparisons |
| ☐ | ☒ | A full description of the statistical parameters including central tendency (e.g. means) or other basic estimates (e.g. regression coefficient) AND variation (e.g. standard deviation) or associated estimates of uncertainty (e.g. confidence intervals) |
| ☐ | ☒ | For null hypothesis testing, the test statistic (e.g. *F*, *t*, *r*) with confidence intervals, effect sizes, degrees of freedom and *P* value noted<br>*Give P values as exact values whenever suitable.* |
| ☒ | ☐ | For Bayesian analysis, information on the choice of priors and Markov chain Monte Carlo settings |
| ☒ | ☐ | For hierarchical and complex designs, identification of the appropriate level for tests and full reporting of outcomes |
| ☒ | ☐ | Estimates of effect sizes (e.g. Cohen's *d*, Pearson's *r*), indicating how they were calculated |

*Our web collection on statistics for biologists contains articles on many of the points above.*

## Software and code

Policy information about availability of computer code

| | |
|---|---|
| Data collection | Single-molecule imaging: a Customized TIRF microscope (IX71 Olympus with EMCCD iXon DU897D, Andor Technology) was used.<br>CryoEM: data were collected on a Titan Krios (ThermoFisher) using SerialEM v3.9.0 at Stanford cEMc and SerialEM v4.0.0beta3 at Janelia Research Campus.<br>Signaling assays: HEK-Blue reporter assays and ELISA assays were read on a SpectraMax Paradigm plate reader (Molecular Devices) using SoftMax Pro v7.1.<br>Surface plasmon resonance data were collected using Biacore T100 control software version 2.0.4. |
| Data analysis | Single-molecule imaging: SLIMfast4C single molecule evaluation software was used and test data sets can be accessed here: https://doi.org/10.5281/zenodo.5712332.<br>CryoEM: cryoSPARC v3.1.0 and v3.2.0 were used for data analysis.<br>Surface plasmon resonance: Biacore T100 Evaluation Software v2.0.4.<br>Structural analysis: Coot v0.9.6; Phenix v1.20.1; UCSF Chimera v1.15; UCSF ChimeraX v1.3; PyMol v2.2.3; qtPISA v2.1.0; ISOLDE v1.3.0; DeepEMhancer version 20210511.<br>Signaling assay analysis and statistical analysis: GraphPad Prism v9.3.1. |

For manuscripts utilizing custom algorithms or software that are central to the research but not yet described in published literature, software must be made available to editors and reviewers. We strongly encourage code deposition in a community repository (e.g. GitHub). See the Nature Portfolio guidelines for submitting code & software for further information.

## Data

Policy information about availability of data

All manuscripts must include a data availability statement. This statement should provide the following information, where applicable:
- Accession codes, unique identifiers, or web links for publicly available datasets
- A description of any restrictions on data availability
- For clinical datasets or third party data, please ensure that the statement adheres to our policy

All single molecule imaging data have been deposited at Zenodo: raw images are at DOI:10.5281/zenodo.6783369 and calibration images are at 10.5281/zenodo.6787325. The CryoEM maps and atomic coordinates for the IL-17RB-IL-25 (2:2), IL-17RB-IL-25 (6:6), IL-17RB-IL-25—-IL-17RA (2:2:2), IL-17RA-IL-17A (2:2), and IL-17RA-IL-17A-IL-17RC (2:2:1) structures have been deposited in the EMDB (EMD-26833, EMD-26834, EMD-26835, EMD-26836, EMD-26837) and PDB (7UWJ, 7UWK, 7UWL, 7UWM, 7UWN), respectively. All other data are available from the corresponding author upon reasonable request.

# Field-specific reporting

Please select the one below that is the best fit for your research. If you are not sure, read the appropriate sections before making your selection.

☒ Life sciences          ☐ Behavioural & social sciences          ☐ Ecological, evolutionary & environmental sciences

For a reference copy of the document with all sections, see nature.com/documents/nr-reporting-summary-flat.pdf

# Life sciences study design

All studies must disclose on these points even when the disclosure is negative.

| | |
|---|---|
| Sample size | Single-molecule experiments: sample sizes were estimated based on extensive statistical analysis of data obtained in a systematic assessment of the single molecule cotracking analyses (Sotolongo Bellon et al., Cell Rep Meth 2, 2022, 100165, DOI:10.1016/j.crmeth.2022.100165). Signaling assays and surface plasmon resonance: no sample size calculations were performed. For signaling and surface plasmon resonance experiments, samples sizes where determined based on previous experience with similar experiments and the sample sizes were adequate based on clear distributions in the data and clear differences among the various conditions. |
| Data exclusions | No data were excluded. |
| Replication | All attempts at replication were successful. Most results were replicated in at least two independent experiments. For some of the single-molecule and SPR conditions only one experiment was performed. The signaling experiments characterizing the IL-17RA KO and IL-17RB KO cell lines were conducted once. The number of independent biological replicates and sample sizes for all data are indicated in the figure legends. |
| Randomization | PBMCs were obtained from multiple donors at random. For in vitro experiments and protein structure determination, independent variables are tightly controlled and therefore covariates are not relevant to those studies. |
| Blinding | Blinding of investigators was not performed or necessary as the readouts of the protein structure determination and in vitro assays are not subjective to investigator bias. |

# Reporting for specific materials, systems and methods

We require information from authors about some types of materials, experimental systems and methods used in many studies. Here, indicate whether each material, system or method listed is relevant to your study. If you are not sure if a list item applies to your research, read the appropriate section before selecting a response.

### Materials & experimental systems

| n/a | Involved in the study |
|---|---|
| ☐ | ☒ Antibodies |
| ☐ | ☒ Eukaryotic cell lines |
| ☒ | ☐ Palaeontology and archaeology |
| ☒ | ☐ Animals and other organisms |
| ☐ | ☒ Human research participants |
| ☒ | ☐ Clinical data |
| ☒ | ☐ Dual use research of concern |

### Methods

| n/a | Involved in the study |
|---|---|
| ☒ | ☐ ChIP-seq |
| ☒ | ☐ Flow cytometry |
| ☒ | ☐ MRI-based neuroimaging |

# Antibodies

| | |
|---|---|
| Antibodies used | Nanobodies: Anti-GFP and anti-ALFAtag (produced in-house). |
| Validation | Nanobodies: Anti-GFP and anti-ALFAtag were previously validated (doi.org/10.1016/j.crmeth.2022.100165 and doi.org/10.1038/s41467-019-12301-7, respectively). Interaction of the in-house produced nanobodies with their target proteins was confirmed by quantitative binding assays. |

# Eukaryotic cell lines

Policy information about cell lines

| | |
|---|---|
| Cell line source(s) | HeLa cells for single-molecule imaging: German Collection of Microorganisms and Cell Cultures GmbH (ACC 57); Expi293F cells (ThermoFisher); Expi293F GnTI- (ThermoFisher); IL-17C HEK-Blue NF-kB Reporter cells (Invivogen). |
| Authentication | HeLa cells (German Collection of Microorganisms and Cell Cultures GmbH), Expi293F cells (ThermoFisher), Expi293F GnTI- (ThermoFisher), and IL-17C HEK-Blue NF-kB Reporter cells (Invivogen) were guaranteed by the suppliers and no additional authentication was performed by the authors of this study. |
| Mycoplasma contamination | HeLa cells for single-molecule imaging were tested negatively for mycoplasma (PCR). Expi293F cells (ThermoFisher) and Expi293F GnTI- cells (ThermoFisher) used for protein production were not tested for mycoplasma contamination by the authors of this study. The IL-17C HEK-Blue NF-kB Reporter cells (Invivogen) were tested for mycoplasma contamination by the manufacturer and no additional testing was performed by the authors of this study. |
| Commonly misidentified lines (See ICLAC register) | None of the cell lines used in this study are commonly misidentified. |

# Human research participants

Policy information about studies involving human research participants

| | |
|---|---|
| Population characteristics | PBMCs were obtained from the Stanford Blood Center from healthy donors. |
| Recruitment | The Stanford Blood Center recruited donors. Written informed consent was obtained from the donors prior to tissue collection. |
| Ethics oversight | The Stanford Blood Center. |

Note that full information on the approval of the study protocol must also be provided in the manuscript.

