## [Peer Review File · Nature]

Manuscript Title: Organizing Structural Principles of the Interleukin-17 Ligand-Receptor Axis

Reviewer Comments & Author Rebuttals

Reviewer Reports on the Initial Version:

Referees' comments:

Referee #1 (Remarks to the Author):

This is an elegant and detailed study by the Garcia group focusing on the architecture of the IL-25-IL-17RB-IL-17RA receptor (with additional comparison to the IL-17A-IL-17RC-IL-17RA complex). Extensive structural studies reveal an unusual 3-dimensional structure described as “tip-to-tip”. Since the IL-17R family is distinct in structure to other cytokine receptor families, defining its receptor interactions from a structural point of view is novel and scientifically important. The work is done very thoroughly with helpful diagrams to convey the fundamental concepts for readers not well versed in structural biology. My concerns are minor, related to some suggested discussion points, overlooked references, and a few issues of accuracy (particularly with respect to IL-17RC). Overall this is a valuable contribution with important implications for understanding cytokine receptor biology and the enigmatic IL-17 family.

comments

1. Legend to Fig 3g - it was unclear what signalling assay was used to draw these conclusions. This should be explained in more detail in text and figure legend
2. For all receptor mutants, it was not clear if the authors verified that equivalent surface expression was preserved – flow cytometry of transfected cells would be sufficient to demonstrate this.
3. NF- κ B activation in HEK-293 cells is an easy model system to measure cytokine signalling, but IL-25 actions in vivo are on mucosal epithelial cells and other downstream pathways are engaged (MAPK, RNA stability). Are similar impairments with these mutants seen in such cell types? (for ex. A549 lung epithelial cells or an intestinal epithelial cell type)
4. Authors “propose that the tip-to-tip architecture of IL-17R subunits provides optimal spacing of IL-17R intracellular domains,” but it is not clear what they think this spacing is really needed for – accommodation of Act1 dimers, or do they envision more than that? It is worth mentioning in the discussion that in the setting of a JAK-STAT activating receptor homodimer (EPOR) it was suggested that distance between transmembrane domains and their associated cytoplasmic tails in the unliganded state is required to prevent bystander JAK-activation (PMID 9974392), whereas here the opposite seems to be the case.
5. In line with the above, an early paper studying IL-17R assembly suggested IL-17RA subunits become more distanced upon ligand binding (PMID 16393951), which would seem to be at least partially in keeping with the findings here. This should be incorporated in discussion
6. Discussion would benefit from implications of structural information for development of anti-cytokine/receptor therapy. Do results make any predictions about design of biologics to treat IL-25 diseases compared to IL-17A-dependent conditions?
7. There is reported intersection between IL-25 and IL-17A signaling – are there implications from

the present results in the setting of shared receptor subunits (PMID 17200411, 18762568).

8. Authors indicate that IL-17RC homodimers do not mediate signalling, but more detail on this point would be helpful for the reader. While the Goepfert et al reference (ref 23) implies that IL-17RC homodimers are signalling-competent, the plurality of data using IL-17RA-ko or IL-17RC-ko mice or in humans (PMID 21350122, 25918342) argue against IL-17RA-independent functions for IL-17. Forced murine IL-17RC homodimers are not signaling competent (PMID 20729198)

9. The authors make incorrect statement that the IL-17RC-SEFIR domain is not required for signal transduction (line 358). The paper cited (ref 17, Hu et al) does not support this statement, but rather shows that the SEFIR is not needed for dimerization with IL-17RC but is needed for signalling (see Fig 4 in that publication). Similar data were reported in PMID 20554964 showing deletions of murine IL-17RC-SEFIR region are refractory to signalling.

Referee #2 (Remarks to the Author):

The activation of IL-17 family receptors induces the production of various of cytokines, which then trigger the inflammation that kills different types of pathogens. Such process is critical for the protection of our bodies from the pathogen's infection. Wilson et. al here presents a series of cryo-EM structures of IL-17 ligand/receptor extracellular complexes, including IL-25/IL-17RB, IL-25/IL-17RB/IL-17RA, IL-17A/IL-17RA and IL-17A/IL-17RA/IL-17RC. While several crystal structures of 2:1 IL-17A/IL-17RA have been reported before, these new cryo-EM structures reveal a tip-to-tip interface between two receptors. Actually, such tip-to-tip interface presents in the crystal of IL-17/IL-17RA complex, but its functional significance has not been realized and experimentally tested before this work. In this work, the authors introduce mutations to IL-17RB to disrupt the tip-to-tip interaction between IL-17RA and IL-17RB, and show that the IL-17RB mutants exhibit the significantly reduced ability in inducing the downstream signaling, suggesting that the tip-to-tip interface is important for the activation of IL-17 family receptors. The authors further propose that the large distance between the membrane proximal regions of two adjacent receptors result from the tip-to-tip interaction may facilitate the recruitment of the downstream effector protein, such as Act1, which explains in part why the tip-to-tip interaction is required for the activation of IL-17 family receptors. But such claim is not supported by any experimental evidence. Overall, the structural and functional works are of good quality. By revealing the structures of different IL-17 receptors/ligands complex in the functional states, this work advances our understanding the structural basis for how IL-17 family receptors are activated. However, there are a few major issues need to be addressed before this work could be published. My specific points are:

(1) In the model based on the cryo-EM map from IL-25/IL-17RB/IL-17RA sample, the authors tentatively assign the densities at both wing regions to IL-17RA. I wonder the certainty and reliability of such modelling. The structures of IL-17RB and IL-17RA are highly similar, and the cryo-EM density for the wing region is resolved at relatively low resolution. I guess the model of IL-17RB might be able to be docked into the density equally well. In addition, the authors show that IL-25/IL-17RB can form even large oligomers via IL-17RB-IL-17RB interaction. The SPR experiments also demonstrate that the IL-25 bound IL-17RB is capable of binding to unliganded immobilized IL-17RB. It is, therefore, possible that this dataset contains compositionally distinct complexes, such as RB-RB-

IL25-RB-RB, RA-RB-IL25-RB-RB, and RA-RB-IL25-RB-RA. The authors need to relax the C2 symmetry during image processing, and perform extensive 3D classification focusing on each wing of the complex with symmetry expansion to separate the particles into different compositional states. This procedure will also allow the authors to improve the density for the wing region and achieve more confident model building. Finally, it is very important that the modelling should be supported by the clear cryo-EM densities of the sidechains of those residues that are unique in either RA or RB.

(2) The authors propose that the spacing between two membrane-proximal domains of the receptors as well as the intracellular region of IL-17RA (longer than that in other receptors of this family) are required for the robust activation of IL-17 family receptors. This is a novel and important insight into the working mechanism of the IL-17 family receptors, but is not supported by any experimental evidence. To test this, the authors could replace the intracellular domain of IL-17RB with that of IL-17RA. If the model is correct, the RB ECD-RA ICD chimeric receptor upon the simulation of IL-25 would have certain ability in triggering downstream signaling in the RA knock-out cell lines, as IL-25 bound RB can form higher order oligomer through tip-to-tip interactions; while the RB ECD-RA ICD chimeric receptor with double mutations in tip (L40A/R46E or D75A/R79E) would not be able to trigger downstream signaling upon IL-25 simulation in both normal and RA knock-out cell lines.

(3) I think it is important to also introduce some mutations to RA to specifically disrupt the tip-to-tip interaction between RA and RB, and test the effect of these mutation in the cell-based experiments. This will provide further evidence to support the functional importance of tip-to-tip interaction.

(4) It would be useful to prepare a figure to compare the structures of IL-17 bound RA and unliganded RA. This will help the readers to understand why the ligand binding is critical for promoting tip-to-tip interaction.

(5) It would be also useful to prepare a supplementary figure to compare the cryo-EM structure of 4:2 IL-17/IL-17RA complexes from this work with the crystal structure of 4:2 IL-17/IL-17RA complexes previously determined.

(6) The authors need to indicate the percentage of the IL-25/IL-17RB particles that form higher order oligomers. Alternatively, the authors could indicate the ratio between dimeric and higher ordered IL-25/IL-17RB particles.

(7) The authors need to calculate the FSC between model and map in order to make sure that the resolution is not over-estimated.

(8) The distance between the two membrane proximal regions of RA in RA-RB-IL25-RB-RA complex is quite difference to that in RA-IL17-RA complex. Could the author provide some discussion for why both complexes can recruit Act1?

(9) Typo in line 162.: should be Y106, not L106.

Referee #3 (Remarks to the Author):

Background:

The IL-17 family of ligands and receptors play important roles in the host defense mechanisms against bacteria, fungi, and helminth infection by inducing cytokines and chemokines, recruiting neutrophils, and modifying T-helper cell differentiation. There are genuine needs to acquire deep mechanistic understanding of how these receptors are activated as they are prominent targets for treating autoimmune disorders.

The IL-17 family consists of six ligands, IL-17A through IL-17F (IL-17E is also known as IL-25) and five receptors (IL-17RA through IL-17RE). The pairing between the ligands and receptors in this family is complicated and not fully understood. For example, most of the ligands in this family are homodimers but IL-17A and IL-17F can also form a heterodimer. On the receptor side, IL-17RA acts as a shared receptor, i.e., ligand-induced signaling requires a ternary complex composed of IL-17RA and a second receptor in the family that varies depending on the cytokine, although the current study showed that the IL-17A/IL-17RA binary complex can also signal. To date, the structural information available for the IL-17 family include: IL-17A or IL-17F homodimer in complex with IL-17RA, IL-17A/F heterodimer in complex with IL-17RA, IL-17F homodimer in complex with IL-17RC. While these structures have been very useful for understanding ligand binding, none of them provided clues for understanding the structural basis of receptor sharing that is critical to receptor activation.

Major findings of this paper:

The most important finding by the authors is a tip-to-tip interaction between two receptor ectodomains that underlies the key organizing principle for the whole IL-17 receptor family, and this interaction is completely unknown for this family. In my opinion, this finding is highly significant because it likely represents a general mode of interaction responsible for the sharing of IL-17RA by other family members and for achieving higher order receptor clustering upon ligand binding as means of receptor control and activation. The authors made the discovery by solving a number of cryo-EM structures including the binary complexes IL-25/IL-17RB and IL-17A/IL-17RA and the ternary complexes IL-17RB/IL-25/IL-17RA and IL-17RA/IL-17A/IL-17RC and supported their structural conclusion with solid functional mutagenesis and single-molecule imaging data.

The most impressive structure is that of the ternary complex comprising IL-25, IL-17RB and IL-17RA at 2:2:2 stoichiometry, showing direct tip-to-tip interaction between IL-17RB and IL-17RA. This mechanism of receptor sharing is very different from that of the gamma chain receptor family in which gamma chain sharing is achieved by concomitant binding of the gamma chain and one of its family members to a cytokine. The authors showed that mutating the tip-to-tip interaction abolished IL-25 signaling. Another finding of conceptual novelty is that ligand binding can structurally prime the receptor ectodomain for the tip-to-tip interaction at a distal site from the ligand. To the best of my knowledge, this represents a novel mechanism of ligand-induced receptor clustering in the field of cytokine receptors. I therefore believe this work represents a major advance in understanding the

signaling mechanism of receptors in the IL-17 family.

I find the study is well designed, the results are crisp, and the paper clearly written. I don't have much to add to this paper except for a few important questions that should be addressed for consistency.

Major points:

1. IL-25 binding to IL-17RB can result in both homodimer (2:2 binary complex) and trimer of dimer (6:6 binary complex). Is the distribution protein concentration dependent and would there be even higher order clusters at higher protein concentration? Can the authors elaborate on the possible function of the 6:6 binary complex given that it is not signaling competent?
2. An inconsistency in the SPR data needs more explanation. IL-17RB showed no binding to immobilized IL-17RA (fig 4e) but not the other way around (extended data fig 6j)?
3. IL-25/IL-17RB can bind to immobilized IL-17RA (fig 4f) but also to immobilized IL-17RB at apparently lower affinity (extended data fig 6f). Would this difference in affinity explain why addition of IL-17RA to IL-25/IL-17RB could compete out the tip-to-tip interaction between two IL-17RBs leading to only 2:2:2 ternary complex, as opposed to a 2:6:6 complex? I also suggest the authors provide K_d values from the SPR data.
4. Fig 5h states that IL-17F cannot signal via IL-17RC homodimer. However, a previous study claimed that the IL-17F/IL-17RC axis is functional under conditions of IL-17RA deficiency (De Luca et al., 2017, Cell Reports 20, 1667–1680). Can the authors elaborate further on the discrepancy?
5. Given the IL-25/IL-17RB can propagate to higher order oligomer (e.g., trimer of dimer) with ligand-induced and tip-to-tip dimerization, I'm surprised that IL-17A/IL-17RA only dimerized via the tip-to-tip interaction, resulting in a 4:2 binary complex (fig 5). Is the 4:2 complex structure just an experimentally trapped complex or actually a signaling competent complex? Is it possible that IL-17A/IL-17RA can form higher order clusters like the IL-25/IL-17RB?

Minor points:

1. In fig 3, the labels for site 1 and site 2 are confusing. For example, L101 is in site 1 and Y92 is in site 2 (figs 3b and 3c), but they are inconsistent with the labeling in fig 3g.
2. I could not find description for C11, B3, G12, B3, B2, A2 in fig 3i.
3. Missing scale bars in figs 1c and 2a.

Author Rebuttals to Initial Comments:

Referee #1 (Remarks to the Author):

This is an elegant and detailed study by the Garcia group focusing on the architecture of the IL-25-IL-17RB-IL-17RA receptor (with additional comparison to the IL-17A-IL-17RC-IL-17RA complex). Extensive structural studies reveal an unusual 3-dimensional structure described as “tip-to-tip”. Since the IL-17R family is distinct in structure to other cytokine receptor families, defining its receptor interactions from a structural point of view is novel and scientifically important. The work is done very thoroughly with helpful diagrams to convey the fundamental concepts for readers not well versed in structural biology. My concerns are minor, related to some suggested discussion points, overlooked references, and a few issues of accuracy (particularly with respect to IL-17RC). Overall this is a valuable contribution with important implications for understanding cytokine receptor biology and the enigmatic IL-17 family.

comments

1. Legend to Fig 3g - it was unclear what signalling assay was used to draw these conclusions. This should be explained in more detail in text and figure legend

These data come from an NF- κ B reporter assay. The figure legend now states this clearly (lines 800–803). We have also updated Figure 3 to include titles above the graphs in panels 3g–i.

2. For all receptor mutants, it was not clear if the authors verified that equivalent surface expression was preserved – flow cytometry of transfected cells would be sufficient to demonstrate this.

We agree that this is an important point, and something we spent considerable time evaluating. We measured surface expression of the receptors by staining their extracellular HA-tags via flow cytometry. Surface expression of WT and mutant receptors was overlapping, and all expressed at significant, but not identical levels (see histogram below).

The rank order of surface expression was:

L40A/R46E > WT = E148R > D75A/R79E.

The rank order of signaling E_{max} was:

WT > E148R > L40A/R46E = D75A/R79E.

Since the large differences in their signaling activities cannot be explained by the small differences in their surface expression levels, our interpretation is that the observed relative activities are due to the mutations, not the changes in surface expression. For example, the double mutants display greatly reduced signaling efficacy, which was not explained by differences in surface expression.

We attempted to normalize surface expression by adjusting the amount of transfected DNA, however surface expression remained very similar regardless. In the revised manuscript Materials and Methods, we have explained this more clearly (please see lines 422–425).

3. NF- κ B activation in HEK-293 cells is an easy model system to measure cytokine signalling, but IL-25 actions in vivo are on mucosal epithelial cells and other downstream pathways are engaged (MAPK, RNA stability). Are similar impairments with these mutants seen in such cell types? (for ex. A549 lung epithelial cells or an intestinal epithelial cell type).

We also performed experiments on bulk PBMCs (Fig. 3h) examining the ability of IL-25 variants to stimulate IL-5 secretion. Using this physiological readout, we observed qualitatively similar impairments between mutants vs. WT as seen with the HEK-293 system. We attempted to examine other proximal signaling pathways (including MAPK/ERK) in A549, HL-60, and HepG2. However, in pilot experiments stimulating cell lines with WT IL-25 and assaying signaling by phosphoflow cytometry, the dynamic range between unstimulated and maximally stimulated was small, which precluded the use of this technique to obtain quantitative parameters such as E_{max} and EC_{50} .

4. Authors “propose that the tip-to-tip architecture of IL-17R subunits provides optimal spacing of IL-17R intracellular domains,” but it is not clear what they think this spacing is really needed for – accommodation of Act1 dimers, or do they envision more than that? It is worth mentioning in the discussion that in the setting of a JAK-STAT activating receptor homodimer (EPOR) it was suggested that distance between transmembrane domains and their associated cytoplasmic tails in the unliganded state is required to prevent bystander JAK-activation (PMID 9974392), whereas here the opposite seems to be the case.

Since the large distance between the receptors is rather unusual, compared to cytokine receptors, we proposed that the spacing is necessary for the recruitment of the Act1 adapter molecules. However, this is entirely speculative and something that will require complex experiments to prove or disprove. While it is known that Act1 forms homotypic interactions with the SEFIR domain of IL-17RA (PMID: 17035243), the stoichiometries of these interactions are not defined, leaving open the possibility for Act1–SEFIR complexes larger than simple heterodimers. In support of that notion, a recent paper (PMID 32696476) using quantitative mass-spec suggests that six Act1 molecules could be recruited to the IL-17RA–IL-17A–IL-17RC heterodimer. We adjusted the text (lines 306–307) to propose that the spacing may be required for recruitment of Act1 ‘molecules’ and not just ‘dimers’:

We don’t think the spacing from EpoR would apply to the IL-17R. First, we have published data that EpoR are not pre-associated on the cell membrane prior to ligand engagement (Wilmes et al., Science 2020), so we think the model the reviewer refers to is probably not physiologically relevant. Second, JAKs are activated by trans-phosphorylation that requires proximity, whereas IL-17R are activated by engagement of a scaffolding protein where the role of proximity is not clear.

5. In line with the above, an early paper studying IL-17R assembly suggested IL-17RA subunits become more distanced upon ligand binding (PMID 16393951), which would seem to be at least partially in keeping with the findings here. This should be incorporated in discussion

We agree that the FRET data in that paper are consistent with our model. We have now addressed this point and cited the paper in the discussion lines 365–368.

6. Discussion would benefit from implications of structural information for development of anti-cytokine/receptor therapy. Do results make any predictions about design of biologics to treat IL-25 diseases compared to IL-17A-dependent conditions?

We believe that the IL-17RA/RB tip-to-tip interface, given the deep pockets in the interface, is an exciting new druggable target for small molecules to achieve IL-25-specific inhibition. We

also think the receptor spacings are important for agonism. We have added the following to the discussion:

Lines 375–382: “The use of anti-IL-17A-specific monoclonal antibody antagonists Secukinumab and Ixekizumab, as well as the anti-IL-17RA monoclonal antibody Brodalumab, which blocks most IL-17 family cytokines, have been successful in the treatment of severe plaque psoriasis³². The structural features of the IL-17R tip-to-tip interactions are compatible with small molecule targeting, for example to inhibit aberrant IL-25 signaling in asthma or psoriasis. The structural and organizing principles of IL-17 family cytokine signaling provides a blueprint for the development of both agonist and antagonist therapeutics.”

7. There is reported intersection between IL-25 and IL-17A signaling – are there implications from the present results in the setting of shared receptor subunits (PMID 17200411, 18762568).

This is an interesting point. PMID 17200411 suggests opposing roles of IL-25 and IL-17A in the pathogenesis of experimental autoimmune encephalomyelitis (EAE), with IL-25 suppressing the disease and IL-17A promoting it. In our study, we identified the tip of IL-17RA as being critical in the formation of both IL-17A- and IL-25-containing complexes. Considering the importance of the IL-17RA tip in IL-17 signaling, it is possible that the opposing roles of IL-17A and IL-25 could be explained, in part, by competition for IL-17RA molecules.

8. Authors indicate that IL-17RC homodimers do not mediate signalling, but more detail on this point would be helpful for the reader. While the Goepfert et al reference (ref 23) implies that IL-17RC homodimers are signalling-competent, the plurality of data using IL-17RA-ko or IL-17RC-ko mice or in humans (PMID 21350122, 25918342) argue against IL-17RA-independent functions for IL-17. Forced murine IL-17RC homodimers are not signaling competent (PMID 20729198)

We have added citations, including PMID 21350122, to the introduction to point out that IL-17RA is required for IL-17A and IL-17F-mediated signaling in mice and humans. This data leads us to question the signaling competence of the IL-17F–IL-17RC homodimers. However, Goepfert et al reference a paper (De Luca et al., 2017, Cell Reports 20, 1667–1680) that shows IL-17F-induced IL-33 expression in the absence of IL-17RA. Rather than dismissing the IL-17F–IL-17RC homodimers, we prefer to cite the De Luca paper and state that the signaling capacity of the IL-17F–IL-17RC homodimers is unclear.

Lines 73–77: “There is also a structure of IL-17F bound to two IL-17RC in a homodimeric complex²³ but whether this entity signals remains unclear. On one hand cellular responses to IL-17A and IL-17F signaling have been shown to require IL-17RA in mice¹⁶ and humans²⁷, on the other hand IL-17F has been shown to upregulate IL-33 in murine cells lacking IL-17RA²⁸.”

We have modified our statements about IL-17RC homodimer signaling in the text (lines 357–358; line 364), and adjusted figure 5h to reflect the ambiguity of the signaling capability of the IL-17F–IL-17RC homodimer.

Figure 5h

9. The authors make incorrect statement that the IL-17RC-SEFIR domain is not required for signal transduction (line 358). The paper cited (ref 17, Hu et al) does not support this statement, but rather shows that the SEFIR is not needed for dimerization with IL-17RC but is needed for signalling (see Fig 4 in that publication). Similar data were reported in PMID 20554964 showing deletions of murine IL-17RC-SEFIR region are refractory to signalling.

The reviewer is correct. We removed the incorrect statement from the discussion section.

Referee #2 (Remarks to the Author):

The activation of IL-17 family receptors induces the production of various of cytokines, which then trigger the inflammation that kills different types of pathogens. Such process is critical for the protection of our bodies from the pathogen's infection. Wilson et. al here presents a series of cryo-EM structures of IL-17 ligand/receptor extracellular complexes, including IL-25/IL-17RB, IL-25/IL-17RB/IL-17RA, IL-17A/IL-17RA and IL-17A/IL-17RA/IL-17RC. While several crystal structures of 2:1 IL-17A/IL-17RA have been reported before, these new cryo-EM structures reveal a tip-to-tip interface between two receptors. Actually, such tip-to-tip interface presents in the crystal of IL-17/IL-17RA complex, but its functional significance has not been realized and experimentally tested before this work. In this work, the authors introduce mutations to IL-17RB to disrupt the tip-to-tip interaction between IL-17RA and IL-17RB, and show that the IL-17RB mutants exhibit the significantly reduced ability in inducing the downstream signaling, suggesting that the tip-to-tip interface is important for the activation of IL-17 family receptors. The authors further propose that the large distance between the membrane proximal regions of two adjacent receptors result from the tip-to-tip interaction may facilitate the recruitment of the downstream effector protein, such as Act1, which explains in part why the tip-to-tip interaction is required for the activation of IL-17 family receptors. But such claim is not supported by any experimental evidence. Overall, the structural and functional works are of good quality. By revealing the structures of different IL-17 receptors/ligands complex in the functional states, this work advances our understanding the structural basis for how IL-17 family receptors are activated. However, there are a few major issues need to be addressed before this work could be published. My specific points are:

(1) In the model based on the cryo-EM map from IL-25/IL-17RB/IL-17RA sample, the authors tentatively assign the densities at both wing regions to IL-17RA. I wonder the certainty and reliability of such modelling. The structures of IL-17RB and IL-17RA are highly similar, and the cryo-EM density for the wing region is resolved at relatively low resolution. I guess the model of IL-17RB might be able to be docked into the density equally well. In addition, the authors show that IL-25/IL-17RB can form even large oligomers via IL-17RB-IL-17RB interaction. The SPR experiments also demonstrate that the IL-25 bound IL-17RB is capable of binding to unliganded immobilized IL-17RB. It is, therefore, possible that this dataset contains compositionally distinct complexes, such as RB-RB-IL25-RB-RB, RA-RB-IL25-RB-RB, and RA-RB-IL25-RB-RA. The authors need to relax the C2 symmetry during image processing, and perform extensive 3D classification focusing on each wing of the complex with symmetry expansion to separate the particles into different compositional states. This procedure will also allow the authors to improve the density for the wing region and achieve more confident model building. Finally, it is very important that the modelling should be supported by the clear cryo-EM densities of the sidechains of those residues that are unique in either RA or RB.

We thank the reviewer for these comments and agree that the requested clarifications are important. We observed no residual higher-order binary complex present in 2D averaging. In our map of the IL-25/IL-17RB/IL-17RA complex, we observed local resolution $<4\text{\AA}$ for the D1 domain of IL-17RA. To ensure that we have determined the entity of the wing correctly, we have analyzed the well-resolved glycosylation at N67 present on the D1 domain of IL-17RA. This glycosylation is not present in IL-17RB, which has an aspartic acid, D39, at this position. Additionally, the location of N67 is on one of the loops at the tip-to-tip interface and is well resolved due to its location nearest to the center of particle mass for RA. This provides definitive placement of IL-17RA into the map. We have added Extended Data Figure 4e, which illustrates this, to the manuscript:

Extended Data Figure 4e, A resolved IL-17RA glycan, N67-NAG, near the tip-to-tip interface distinguishes IL-17RA from IL-17RB in the wing of the ternary complex. Models of IL-17RB (red and grey) and IL-17RA (purple) are docked into the cryoEM density of the IL-25 ternary complex map.

We agree that a focused 3D classification approach would improve confidence in the model. Using the N67-NAG as a fiducial, we determined that IL-17RA is the component in the “wing” in each of the five particle classes (87,775-90,652 particles per class) and visualized this density clearly:

(2) The authors propose that the spacing between two membrane-proximal domains of the receptors as well as the intracellular region of IL-17RA (longer than that in other receptors of this family) are required for the robust activation of IL-17 family receptors. This is a novel and important insight into the working mechanism of the IL-17 family receptors, but is not supported by any experimental evidence. To test this, the authors could replace the intracellular domain of IL-17RB with that of IL-17RA. If the model is correct, the RB ECD-RA ICD chimeric receptor upon the simulation of IL-25 would have certain ability in triggering downstream signaling in the RA knock-out cell lines, as IL-25 bound RB can form higher order oligomer through tip-to-tip interactions; while the RB ECD-RA ICD chimeric receptor with double mutations in tip (L40A/R46E or D75A/R79E) would not be able to trigger downstream signaling upon IL-25 simulation in both normal and RA knock-out cell lines.

Given that other systems such as cytokines and RTK assemble receptor signaling complexes that are generally within close proximity to allow trans-phosphorylation, we speculated on a possible reason for the rather unusually wide spacing between the 17R dimers. One hypothesis is that Act1 acts as a scaffold that requires accommodation by a certain spacing between the receptors. Proving this definitively will be very challenging. Experiments using a molecular ruler, such as we applied to EPO receptor (Mohan et al., Science 2019) could provide some resolution to this question. However, these experiments are significantly outside the scope of the current manuscript.

We appreciate the interesting idea for the ICD swapping experiment. We discussed this extensively, but for several key reasons we do not think it would conclusively answer the spacing question for the following reasons:

- In both the IL-25 and IL-17A/F systems, there is likely some obligate requirement for the SEFIR domain of IL-17RB (Rickel *et al J. Immunol.* 181, 4299–4310 (2008)) or IL-17RC (Hu *et al. J. Immunol.* 184, 4307–4316 (2010)) for signaling, so it is likely that this chimera would not signal. Furthermore, our own knockout data in Figure 3i confirm the findings of Rickel et al. that both IL-17RA and IL-17RB are required for signaling.
- The physiological relevance of the IL-17RB oligomeric assembly is not known. In total the population of clustered higher-order IL-17RB on a cell in our TIRFM data is less than 10%, and form transiently so the IL-17RB ECD/IL-17RA ICD molecules within an IL-17RB array would likely not signal strongly enough for a robust signal. Please see lines 219–221: “Single-molecule intensity analyses suggested that higher-order clusters are formed transiently and at a rather low level (<10% of the co-diffusing fraction, Extended Data Fig. 5e).”
- In our TIRFM data for the IL-17RB tip mutants, we did not see a level of RB–RB co-localization reduction comparable to the very strong effect the mutations have on RA–RB (Figure 4b, c). Thus, the RB mutations would likely not fully disrupt the higher-order IL-17RB arrays enough to disrupt signaling.

(3) I think it is important to also introduce some mutations to RA to specifically disrupt the tip-to-tip interaction between RA and RB, and test the effect of these mutation in the cell-based experiments. This will provide further evidence to support the functional importance of tip-to-tip interaction.

We provided extensive validation data on the RA–RB tip-to-tip interface using IL-17RB tip mutants in cell signaling (Fig. 3j), single-molecule imaging (Fig. 4b), and SPR experiments (Extended Data Fig. 6b, c).

We agree that validating the same interface using IL-17RA tip mutants would further confirm this interaction, but we encountered technical issues in executing these experiments. We generated IL-17RA tip mutants and attempted to assess their effects on signaling in IL-17RA KO cells, using our NF-κB reporter assay. Unfortunately, despite repeated attempts, the results of our assays were inconclusive due to basal signaling, and loss of cytokine responsiveness, of the transfected IL-17RA variants. In these experiments, the re-introduced mutant IL-17RA variants signal aberrantly, without the addition of cytokine. Unfortunately, IL-17RA does not tolerate mutations as well as IL-17RB. The data for our assays are below:

(4) It would be useful to prepare a figure to compare the structures of IL-17 bound RA and unliganded RA. This will help the readers to understand why the ligand binding is critical for promoting tip-to-tip interaction.

We added a comparison of the unliganded and liganded IL-17A structures to Extended Data Figure 8, as panel b.

Extended Data Figure 8 | IL-17RA structural overlay. a, IL-17RA from the cryoEM analysis superimposed with the liganded 3JVF crystal structure, 5NAN crystal structure, and 4HSA crystal structure, in purple, green, blue, and pink, respectively. b, IL-17RA from the cryoEM analysis superimposed with the unliganded 5N9B crystal structure, in purple and grey respectively. Regions where the unliganded structure has

unresolved loops are colored in yellow, while regions where the unliganded structure has less ordered secondary structure are colored in red.

(5) It would be also useful to prepare a supplementary figure to compare the cryo-EM structure of 4:2 IL-17/IL-17RA complexes from this work with the crystal structure of 4:2 IL-17/IL-17RA complexes previously determined.

Extended Data Figure 8 (now 8a) compares our IL-17RA complex to the available crystal structures (please see above).

(6) The authors need to indicate the percentage of the IL-25/IL-17RB particles that form higher order oligomers. Alternatively, the authors could indicate the ratio between dimeric and higher ordered IL-25/IL-17RB particles.

In our live cell single-molecule imaging we observe that <10% of the co-diffusing fraction form higher-order clusters. We observe that approximately 18% of our well aligning particles are part of higher-order structures in the cryoEM data. The higher value is perhaps due to concentration differences as well as affinity differentials between extracellular domain and full-length constructs. We have added to our statement regarding the live cell percentage with the following:

Lines 219–223: “Single-molecule intensity analyses suggested that higher-order clusters are formed as observed in vitro, yet at a rather low level (<10% of the co-diffusing fraction, Extended Data Fig. 5e). This low level of higher-order clustering is consistent with the values seen in our cryoEM samples, where approximately 18% of particles form higher order clusters.”

(7) The authors need to calculate the FSC between model and map in order to make sure that the resolution is not over-estimated.

We added map-model FSC curves to each of the supplemental data processing figures.

(8) The distance between the two membrane proximal regions of RA in RA-RB-IL25-RB-RA complex is quite different to that in RA-IL17-RA complex. Could the author provide some discussion for why both complexes can recruit Act1?

While the spacing between the two membrane proximal regions of RA in these two structures is quite different, the spacing between the homo- and hetero- IL-17RA interactions is essentially the same (~93 Å) as noted in Figure 5g and our schematic below. The conserved spacing between either the homo- or hetero-RA tip-to-tip complexes may play a role in the recruitment of Act1, though this is speculative.

We have adjusted the discussion to clarify our discussion of homo- and hetero-RA interactions as follows:

Lines 368–374: “We propose that the tip-to-tip architecture provides the optimal arrangement of IL-17R intracellular domains required for homotypic IL-17R-SEFIR–Act1 interactions and signal transduction (Fig. 5h). It is clear the intracellular region of IL-17RA plays a key role, whether this is in the context of homo- or hetero-tip-to-tip interactions. Further studies are required to determine how the tip-to-tip architecture tunes the spacing and geometry of IL-17–IL-17R complexes and Act1 signaling.”

(9) Typo in line 162.: should be Y106, not L106.

We have corrected this typo.

Referee #3 (Remarks to the Author):

Background:

The IL-17 family of ligands and receptors play important roles in the host defense mechanisms against bacteria, fungi, and helminth infection by inducing cytokines and chemokines, recruiting neutrophils, and modifying T-helper cell differentiation. There are genuine needs to acquire deep mechanistic understanding of how these receptors are activated as they are prominent targets for treating autoimmune disorders.

The IL-17 family consists of six ligands, IL-17A through IL-17F (IL-17E is also known as IL-25) and five receptors (IL-17RA through IL-17RE). The pairing between the ligands and receptors in this family is complicated and not fully understood. For example, most of the ligands in this family are homodimers but IL-17A and IL-17F can also form a heterodimer. On the receptor side, IL-17RA acts as a shared receptor, i.e., ligand-induced signaling requires a ternary complex composed of IL-17RA and a second receptor in the family that varies depending on the cytokine, although the current study showed that the IL-17A/IL-17RA binary complex can also signal. To date, the structural information available for the IL-17 family include: IL-17A or IL-17F homodimer in complex with IL-17RA, IL-17A/F heterodimer in complex with IL-17RA, IL-17F homodimer in complex with IL-17RC. While these structures have been very useful for understanding ligand binding, none of them provided clues for understanding the structural basis of receptor sharing that is critical to receptor activation.

Major findings of this paper:

The most important finding by the authors is a tip-to-tip interaction between two receptor ectodomains that underlies the key organizing principle for the whole IL-17 receptor family, and this interaction is completely unknown for this family. In my opinion, this finding is highly significant because it likely represents a general mode of interaction responsible for the sharing of IL-17RA by other family members and for achieving higher order receptor clustering upon ligand binding as means of receptor control and activation. The authors made the discovery by solving a number of cryo-EM structures including the binary complexes IL-25/IL-17RB and IL-17A/IL-17RA and the ternary complexes IL-17RB/IL-25/IL-17RA and IL-17RA/IL-17A/IL-17RC and supported their structural conclusion with solid functional mutagenesis and single-molecule imaging data.

The most impressive structure is that of the ternary complex comprising IL-25, IL-17RB and IL-17RA at 2:2:2 stoichiometry, showing direct tip-to-tip interaction between IL-17RB and IL-17RA. This mechanism of receptor sharing is very different from that of the gamma chain receptor family in which gamma chain sharing is achieved by concomitant binding of the gamma chain and one of its family members to a cytokine. The authors showed that mutating the tip-to-tip interaction abolished IL-25 signaling. Another finding of conceptual novelty is that ligand binding can structurally prime the receptor ectodomain for the tip-to-tip interaction at a distal site from the ligand. To the best of my knowledge, this represents a novel mechanism of ligand-induced receptor clustering in the field of cytokine receptors. I therefore believe this work represents a major advance in understanding the signaling mechanism of receptors in the IL-17 family.

I find the study is well designed, the results are crisp, and the paper clearly written. I don't have much to add to this paper except for a few important questions that should be addressed for consistency.

Major points:

1. IL-25 binding to IL-17RB can result in both homodimer (2:2 binary complex) and trimer of dimer (6:6 binary complex). Is the distribution protein concentration dependent and would there be even higher order clusters at higher protein concentration? Can the authors elaborate on the possible function of the 6:6 binary complex given that it is not signaling competent?

In our live cell imaging experiments, we see <10% of the co-tracking fraction in higher order clusters while in our cryoEM samples we see ~18% of the population that could be resolved in 2D averaging to be in higher order clustering (see lines 219–223). This could be an effect of a higher concentration present in the cryoEM samples, though comparisons are not informative since this is purified ECD, as opposed to the full-length protein of the live cell imaging. We speculate in the discussion that IL-17A signaling complexes are determined “as a function of ligand concentration and receptor expression” and we have now expanded this statement to include IL-25 complexes to conclude this paragraph in the discussion (please see lines 357–361).

We speculate that the putative non-signaling 6:6 binary complexes could function to sequester IL-25 to modulate IL-25 signaling. However, the physiological relevance of the IL-17RB assembly is not known without further experimentation.

2. An inconsistency in the SPR data needs more explanation. IL-17RB showed no binding to immobilized IL-17RA (fig 4e) but not the other way around (extended data fig 6j)?

The reviewer is correct, and this is something we often see in SPR, where weak interactions can be orientation sensitive due to slight conformational distortions when immobilized on the chip. Despite there being a weak IL-17RB–IL-17RA interaction detected with SPR we did not detect an IL-25-independent IL-17RB–IL-17RA interaction on the cell surface using single-molecule imaging (Fig. 4a, b). We added the following statement to the text:

Lines 266–269: “Despite the weak apo-IL-17RB–apo-IL-17RA interaction detected by SPR, we did not detect an IL-25-independent IL-17RB–IL-17RA interaction using single-molecule live cell imaging (Fig. 4b).”

3. IL-25/IL-17RB can bind to immobilized IL-17RA (fig 4f) but also to immobilized IL-17RB at apparently lower affinity (extended data fig 6f). Would this difference in affinity explain why addition of IL-17RA to IL-25/IL-17RB could compete out the tip-to-tip interaction between two IL-17RBs leading to only 2:2:2 ternary complex, as opposed to a 2:6:6 complex? I also suggest the authors provide K_d values from the SPR data.

The reviewer’s suggestion is formally possible. Since the IL-17RA–IL-17RB affinity is higher than the IL-17RB–IL-17RB affinity it is possible that IL-17RA out competed IL-17RB for all the IL-17RB binding sites.

We added K_d values for the SPR data to Fig. 4 and Extended Data Fig.6. While we detected IL-25–IL-17RB binding to immobilized IL-17RB that was lower affinity than binding to IL-17RA, the SPR curves were difficult to fit to standard binding isotherm models, likely due to the formation of higher-order complexes on the chip. Therefore, we did not report a K_d for the IL-25–IL-17RB–IL-17RB interaction. We have now added the K_ds for the IL-25–IL-17RB–IL-17RA interaction and the IL-17RA–IL-17RB interaction to the figures and the results section (lines 264–266).

4. Fig 5h states that IL-17F cannot signal via IL-17RC homodimer. However, a previous study claimed that the IL-17F/IL-17RC axis is functional under conditions of IL-17RA deficiency (De Luca et al., 2017, Cell Reports 20, 1667–1680). Can the authors elaborate further on the discrepancy?

Reviewer #1 also raised this question, so our response is the same. We are skeptical whether the IL-17RC homodimer signals in the absence of IL-17RA, as is claimed in the De Luca paper. We have added citations, including PMID 21350122, to the introduction to point out that IL-17RA is required for IL-17A- and IL-17F-mediated signaling in mice and humans. However, rather than dismissing the IL-17F–IL-17RC homodimers, we prefer to state that the competence of the IL-17F–IL-17RC homodimers is unclear. We have modified the text and Figure 5h:

Lines 73–77: “There is also a structure of IL-17F bound to two IL-17RC in a homodimeric complex²³ but whether this entity signals remains unclear. On one hand cellular responses to IL-17A and IL-17F signaling have been shown to require IL-17RA in mice¹⁶ and humans,²⁷ on the other hand IL-17F has been shown to upregulate IL-33 in murine cells lacking IL-17RA²⁸.”

We have modified our statements about IL-17RC homodimer signaling in the text (Lines 357-359; line 364), and adjusted figure 5h to reflect the ambiguity of the signaling competence of the IL-17F–IL-17RC homodimer.

Figure 5h

5. Given the IL-25/IL-17RB can propagate to higher order oligomer (e.g., trimer of dimer) with ligand-induced and tip-to-tip dimerization, I'm surprised that IL-17A/IL-17RA only dimerized via the tip-to-tip interaction, resulting in a 4:2 binary complex (fig 5). Is the 4:2 complex structure just an experimentally trapped complex or actually a signaling competent complex? Is it possible that IL-17A/IL-17RA can form higher order clusters like the IL-25/IL-17RB?

Based on our data, and the various additional IL-17A, A/F, and F crystal structures, we think that binding of IL-17RA to two copies of IL-17A, IL-17A/F, or IL-17F is not favored. Therefore, we do not think, nor did we observe in our cryoEM data that it would form higher order clusters as seen with IL-17RB, which was able to bind at two faces of the cytokine, in addition to tip-to-tip.

In addition, we have clarified the manuscript that IL-17A/IL-17RA 4:2 binary complex is not signaling competent (Figure 5h; please see above) – because it requires IL-17RC for signaling (Hu *et al. J. Immunol.* 184, 4307–4316 (2010)).

Therefore, we believe that the 4:2 complex observed in the crystal structures and the cryoEM structure, is a physiologically relevant complex, albeit a non-signaling, potentially “primed” complex on the cell surface.

Minor points:

1. In fig 3, the labels for site 1 and site 2 are confusing. For example, L101 is in site 1 and Y92 is in site 2 (figs 3b and 3c), but they are inconsistent with the labeling in fig 3g.

There was an error in the labeling in figure 3g. We corrected the labeling in the figure.

2. I could not find description for C11, B3, G12, B3, B2, A2 in fig 3i.

These are the identities for the different clones for the IL-17RA and IL-17RB knockouts. We updated the figure legend to specify that these IDs represent different clones.

Lines 807–808: “*C11, B3, G12, B3, B2, A2* represent different clones.”

3. Missing scale bars in figs 1c and 2a.

We added the scale bars to figures 1c and 2a and noted their lengths in both figure legends. Please see lines 777–778 and 785–786.

Reviewer Reports on the First Revision:

Referees' comments:

Referee #1 (Remarks to the Author):

The authors have addressed the reviewers' comments in a highly satisfactory manner. I have no further concerns, and congratulate them on a beautiful and important story.

Referee #2 (Remarks to the Author):

The authors did not perform the ICD swap experiments I suggested, but they provide reasonable justifications for why such experiments cannot provide more evidences to further support the important role of spacing between two IL-17RA-ICDs in initiating IL17 downstream signaling. This requires further investigation in the future. The authors have addressed most of my other points in the revised manuscript. The structural and functional results present in the current manuscript are already very valuable. There is no need to delay the publication. I would support the publication of the current version of manuscript at Nature.

Referee #3 (Remarks to the Author):

The authors have address most of the important issues I had. As I mentioned in my original review, this finding is highly significant for understanding the signaling mechanism of receptors in the IL-17 family. I believe the revised paper is in a good shape for publication.